# TTLL1 and TTLL4 polyglutamylases are required for the neurodegenerative phenotypes in *pcd* mice

Hui-Yuan Wu[1,2], Yongqi Rong[1⊚], Parmil K. Bansal[1⊚], Peng Wei[1⊚], Hong Guo[1], James I. Morgan[1]*

**1** Department of Developmental Neurobiology, St. Jude Children's Research Hospital, Memphis, Tennessee, United States of America, **2** School of Pharmaceutical Science and Technology, Tianjin University, Tianjin, China

⊚ These authors contributed equally to this work.

* jim.morgan@stjude.org

**Data Availability Statement:** All relevant data are within the manuscript and its Supporting Information files.

## Abstract

Polyglutamylation is a dynamic posttranslational modification where glutamate residues are added to substrate proteins by 8 tubulin tyrosine ligase-like (TTLL) family members (writers) and removed by the 6 member Nna1/CCP family of carboxypeptidases (erasers). Genetic disruption of polyglutamylation leading to hyperglutamylation causes neurodegenerative phenotypes in humans and animal models; the best characterized being the *Purkinje cell degeneration* (*pcd*) mouse, a mutant of the gene encoding Nna1/CCP1, the prototypic eraser. Emphasizing the functional importance of the balance between glutamate addition and elimination, loss of TTLL1 prevents Purkinje cell degeneration in *pcd*. However, whether *Ttll1* loss protects other vulnerable neurons in *pcd*, or if elimination of other TTLLs provides protection is largely unknown. Here using a mouse genetic rescue strategy, we characterized the contribution of *Ttll1*, *4*, *5*, *7*, or *11* to the degenerative phenotypes in cerebellum, olfactory bulb and retinae of *pcd* mutants. *Ttll1* deficiency attenuates Purkinje cell loss and function and reduces olfactory bulb mitral cell death and retinal photoreceptor degeneration. Moreover, degeneration of photoreceptors in *pcd* is preceded by impaired rhodopsin trafficking to the rod outer segment and likely represents the causal defect leading to degeneration as this too is rescued by elimination of TTLL1. Although TTLLs have similar catalytic properties on model substrates and several are highly expressed in Purkinje cells (e.g. TTLL5 and 7), besides TTLL1 only TTLL4 deficiency attenuated degeneration of Purkinje and mitral cells in *pcd*. Additionally, TTLL4 loss partially rescued photoreceptor degeneration and impaired rhodopsin trafficking. Despite their common properties, the polyglutamylation profile changes promoted by TTLL1 and TTLL4 deficiencies in *pcd* mice are very different. We also report that loss of anabolic TTLL5 synergizes with loss of catabolic Nna1/CCP1 to promote photoreceptor degeneration. Finally, male infertility in *pcd* is not rescued by loss of any *Ttll*. These data provide insight into the complexity of polyglutamate homeostasis and function *in vivo* and potential routes to ameliorate disorders caused by disrupted polyglutamylation.

**Funding:** This work was supported in part by the NCI (National Cancer Institute) Cancer Center Support Grant CA 21765, NIH (National Institutes of Health) grant NS051537 to J.I.M. and ALSAC (American Lebanese Syrian Associated Charities) to J.I.M. The funders had no role in study design, data collection and analysis, decision to publish, or preparation of the manuscript.

**Competing interests:** The authors have declared that no competing interests exist.

## Author summary

Polyglutamylation is a process that modifies proteins with a degradable side chain of glutamate residues. The enzymes that catalyze the addition and removal of the side chain are tubulin tyrosine ligase like (TTLL) members and 6-member cytosolic carboxypeptidase (CCP) family, respectively. Mutations of Nna1/CCP1 cause severe neurodegeneration across species, including human, while the best characterized is the *Purkinje cell degeneration* (*pcd*) mouse, which exhibits progressive loss of cerebellar Purkinje cells, olfactory bulb mitral cells and retinal photoreceptors in addition to male infertility. Although Nna1 can metabolize products of multiple TTLLs *in vitro*, it remains largely unknown how these TTLLs contribute to the *pcd* phenotypes. To this end, we systematically examined whether null mutation of *Ttll1*, *4*, *5*, *7*, or *11* can rescue the phenotypes of *pcd* mice. We showed that *Ttll1* deficiency rescues Purkinje cell loss and function, and also preserves olfactory bulb mitral cells and retinal photoreceptors. Elimination of TTLL4, but not other TTLLs, spares Purkinje and mitral cells and partially rescues photoreceptor degeneration. However, only loss of *Ttll1* but not *Ttll4* corrected the excessive tubulin polyglutamylation in *pcd* cerebellum and loss of none of the *Ttll*s rescued male infertility. This study pointed to a potential therapeutic opportunity.

## Introduction

Protein polyglutamylation and de-glutamylation, which are catalyzed, respectively, by tubulin tyrosine ligase-like (TTLL) and cytosolic carboxypeptidase (CCP) enzymes, contribute to diverse biological processes and their mutation underlie rare neurodegenerative disorders. Polyglutamylation consists of the successive enzymatic addition of glutamate molecules to an acceptor glutamic acid within a substrate protein. During polyglutamate chain initiation, the first glutamate is added to the gamma-carboxyl group of a glutamic acid residue. Subsequently, polymer elongation occurs through glutamate addition to the alpha-carboxyl group of this initiating glutamic acid, and continues through iterative alpha linked additions [1]. This is a dynamic process where glutamate polymerization is catalyzed by several TTLL family members, whereas degradation of the polymer is catalyzed by the 6-member CCP family [1–3]. A range of proteins undergo polyglutamylation [4], with tubulin being the first identified and most extensively studied [1,5,6]. Indeed, the substrate specificities of TTLLs and CCPs were mainly determined with tubulin as substrate [1–3,7,8]. TTLL4 and TTLL5 catalyze the formation of initiating γ-carboxyl linkages with preferences for β- and α-tubulin, respectively. TTLL1 and 11 catalyze the formation of α-carboxyl chain-elongating linkages primarily in α-tubulin. TTLL7 catalyzes both initiation and elongation of the polyglutamate side chain of β-tubulin [1,9]. CCP5 uniquely catalyzes removal of the gamma-carboxyl linked branch point glutamate of tubulins [2,10]. The other 5 CCP family members all cleave α-carboxyl linked glutamate residues on tubulin [2,3] although they can be distinguished to some extent by their kinetic properties and synthetic substrate preferences *in vitro* [10,11] and their non-redundant function *in vivo* [11–14].

 Despite its discovery 30 years ago, only relatively recently has a link been established between disrupted polyglutamylation and neurodegeneration *in vivo*. This emanated from the recognition that the zinc metallocarboxypeptidase, Nna1 (a.k.a. CCP1) [15], which is mutated in the recessive *Purkinje cell degeneration* (*pcd*) mouse [16] is a deglutamylase capable of degrading polyglutamate side chains [2,7,8]. In addition to male infertility, *pcd* mice are

characterized by progressive and selective neurodegeneration of cerebellar Purkinje cells, olfactory bulb mitral cells, thalamic neurons, spinal motor neurons and retinal photoreceptors [17,18]. In parallel, tubulin polyglutamylation levels are increased in the central nervous system and testis of *pcd* mice [2,7,10]. Mutations of *Agtpbp1*, the gene encoding Nna1/CCP1, have also been shown to cause lower motor neuron degeneration in sheep [19] and in humans cause an infantile-onset, progressive, and severe neurodegeneration [18,20,21].

A picture is emerging of complex relationships amongst the anabolic and catabolic enzymes involved in polyglutamylation, their substrates and neurodegenerative phenotypes. Elevated tubulin glutamylation, as observed in *pcd* mice, is associated with neurodegeneration. However, the converse is not true as loss of TTLL1 function results in reduced glutamylation but no neurodegeneration [6] and targeted over-expression of Nna1, CCP4 or CCP6 in Purkinje cells, which degrades polyglutamate chains does not result in their degeneration [11]. These data imply it is longer, rather than shorter, polyglutamate chains that are deleterious to neurons. The role of the balance between anabolism and catabolism of glutamate chains in neurodegeneration is exemplified by the finding that loss of TTLL1 function, which reduces polyglutamylation, rescues Purkinje cell degeneration in *pcd* mice [7,22].

Many TTLLs are expressed in brain, some at levels similar to, or higher than TTLL1 [5,23,24]. This begs the question of whether additional TTLLs also contribute to neurodegeneration in *pcd* mice. For example, as TTLL1 mediates elongation of glutamate chains on α-tubulin it might be anticipated that loss of an initiator TTLL for this tubulin isoform such as TTLL5 might produce a similar outcome. Here we test null alleles of different TTLLs for their ability to modify tubulin polyglutamylation and rescue degeneration of retinal photoreceptors, olfactory bulb mitral cells and cerebellar Purkinje neurons as well as testicular structure and function in *pcd* mice.

## Results

### Polyglutamate homeostasis and *Ttll* expression patterns

Glutamylation is a complex balance between glutamate addition by the TTLL family of enzymes and their elimination by the CCP family of carboxypeptidases ([1,2,7,8,14] and Fig 1A). Thus, glutamylation is dependent both on the levels of the various enzymes in particular cells and tissues, their catalytic properties (e.g. initiators or elongators) and preferences for different substrates (e.g. α- vs β-tubulin). The glutamylation homeostatic process is depicted in Fig 1A for α- and β-tubulins. It should be borne in mind that substrate preferences have been largely established *in vitro* and less is known about their enzymatic properties and specificities *in vivo*.

To determine which TTLL glutamylases might contribute to the *pcd* phenotype, we examined their expression by quantitative RT-PCR in cerebellum, olfactory bulb and testis which have abnormalities in *pcd* mice [25] as well as cerebral cortex, kidney and liver that have no reported deficiencies [17,25] (Fig 1B). In general, expression of *Ttll*s was highest in testis followed by the three brain regions and lowest levels were observed in kidney and liver. Of note, *Agtpbp1*, the gene encoding Nna1/CCP1 and is mutated in the *pcd* mouse, is highly expressed in brain compared to non-neural tissues and is even present at high levels in cerebral cortex which has no reported degeneration.

### Loss of *Ttll1* or *Ttll4*, but not *Ttll5*, *7*, or *11*, attenuates Purkinje cell degeneration in *pcd* mice

TTLL1, 2, 4, 5, 6, 7, 9, 11 and 13 are proven glutamylases whereas TTLL3, 8 and 10 are glycylases and TTLL12 has no known enzymatic activity [1,26,27]. Of the authenticated

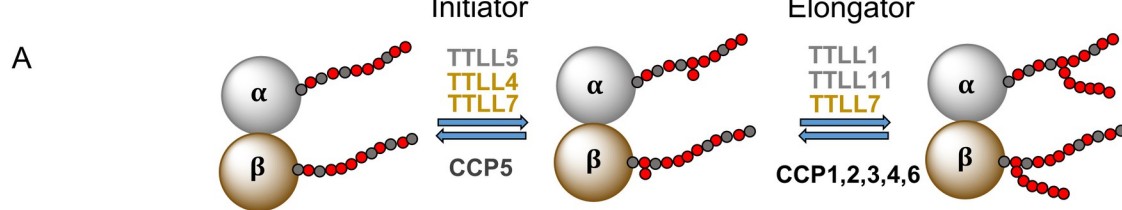

| *Gene* | CB | CX | OB | TS | KD | LV |
|---|---|---|---|---|---|---|
| *Agtpbp1* | 3429 ±23.9 | 2473 ±5.2 | 1405 ±36.2 | 5405 ±465.4 | 360 ±7.4 | 119 ±4.6 |
| *Ttll1* | 467 ±20.8 | 798 ±10.8 | 772 ±11.5 | 1188 ±8.2 | 195 ±2.4 | 74 ±2.1 |
| *Ttll2* | 2 ±0.0 | 1 ±0.1 | 8 ±0.1 | 613 ±85.3 | 56 ±7.1 | 2 ±0.4 |
| *Ttll3* | 864 ±13.7 | 146 ±0.1 | 209 ±6.7 | 5331 ±316.2 | 69 ±4.6 | 21 ±0.8 |
| *Ttll4* | 298 ±20.1 | 105 ±1.0 | 176 ±3.1 | 2678 ±31.9 | 299 ±7.8 | 147 ±3.0 |
| *Ttll5* | 1532 ±31.8 | 347 ±6.1 | 349 ±3.3 | 6369 ±119.1 | 190 ±4.2 | 192 ±7.8 |
| *Ttll6* | 6 ±0.6 | 3 ±0.1 | 4 ±0.4 | 4000 ±350.9 | 1 ±0.0 | 0 ±0.0 |
| *Ttll7* | 6809 ±92.3 | 7777 ±60.5 | 5922 ±62.4 | 7485 ±561.7 | 107 ±1.5 | 91 ±2.2 |
| *Ttll8* | 37 ±1.3 | 29 ±0.3 | 45 ±1.5 | 14688 ±2043.0 | 4 ±0.2 | 3 ±0.1 |
| *Ttll9* | 134 ±3.4 | 36 ±0.7 | 16 ±0.7 | 21409 ±3001.7 | 17 ±1.2 | 2 ±0.2 |
| *Ttll10* | 2 ±0.3 | 1 ±0.1 | 2 ±0.5 | 19478 ±2138.8 | 326 ±13.5 | 6 ±0.4 |
| *Ttll11* | 636 ±9.8 | 1109 ±6.6 | 445 ±0.5 | 3226 ±376.6 | 7 ±0.2 | 5 ±0.3 |
| *Ttll12* | 348 ±11.1 | 227 ±1.6 | 289 ±6.5 | 695 ±68.7 | 404 ±11.4 | 249 ±2.6 |
| *Ttll13* | 1 ±0.1 | 0 ±0.0 | 1 ±0.1 | 4582 ±488.2 | 2 ±0.1 | 2 ±0.1 |

**Fig 1. Expression and properties of *Ttll* polyglutamylases.** (A) Schematic representation of anabolic and catabolic enzymes metabolizing polyglutamate chains of tubulin. Initiator and elongator TTLLs differentially prefer α-tubulin (gray) or β-tubulin (brown) [1,5]. CCP5 uniquely cleaves the branching point glutamate, whereas CCP1, 2, 3, 4 and 6 degrade the α-carboxyl linked glutamates in the chain. (B) Expression profiles for *Ttlls* and *Agtpbp1* (gene encoding Nna1/CCP1). Expression was determined using qRT-PCR on total RNA from adult cerebellum (CB), cerebral cortex (CX), olfactory bulb (OB), testis (TS), kidney (KD), and liver (LV). Data represents mean and SEM of transcript copy numbers/ng total RNA from 3 independent mice. Authentic polyglutamylases are indicated in black whereas *Ttll3*, *8* and *10* (blue) are glycylases [1] and *Ttll12* (red) is of unknown activity [1].

glutamylases, expression of *Ttll2*, *6* and *13* are at the limit of detection in neural tissues and *Ttll9* has relatively low expression (Fig 1B). Reviewing the Allen Mouse Brain Atlas and published literature [5,24,28] *Ttll1*, *4*, *5*, *7*, and *11* are expressed at various levels in cerebellar Purkinje cells and mitral cells of the olfactory bulb that degenerate in *pcd* mice whereas *Ttll9* has marginal or no expression in either neuronal type and no null strain was available to us. Therefore, we focused on *Ttll1*, *4*, *5*, *7* and *11* and crossed null alleles of each (S1 Fig) onto a *pcd* background. Absence of transcripts in all null strains was confirmed by RT-PCR of total RNA from various tissues (Figs 2A, 2B and S2).

All strains of mice were first tested for locomotor performance on an accelerating rota-rod and subsequently sacrificed and survival of Purkinje cells and levels of polyglutamylation in cerebellum determined by immunofluorescence and western blotting, respectively. In *pcd* mice, only loss of *Ttll1* or *Ttll4* restored locomotor activity to levels comparable to wild-type mice (Fig 2C and 2D). No significant improvement of rota-rod performance was seen with null alleles of *Ttll5*, *7* or *11* in *pcd* mice (S2 Fig).

Degeneration of Purkinje cells in *pcd* begins around day 17 after birth and progresses over several months and affects some lobes before others [17,25]. Purkinje cell integrity was assessed using immunofluorescence with an anti-calbindin antibody (Fig 3). Low power images reveal that cerebellum of *pcd* (Fig 3B) is smaller than that of wild-type (Fig 3A) or

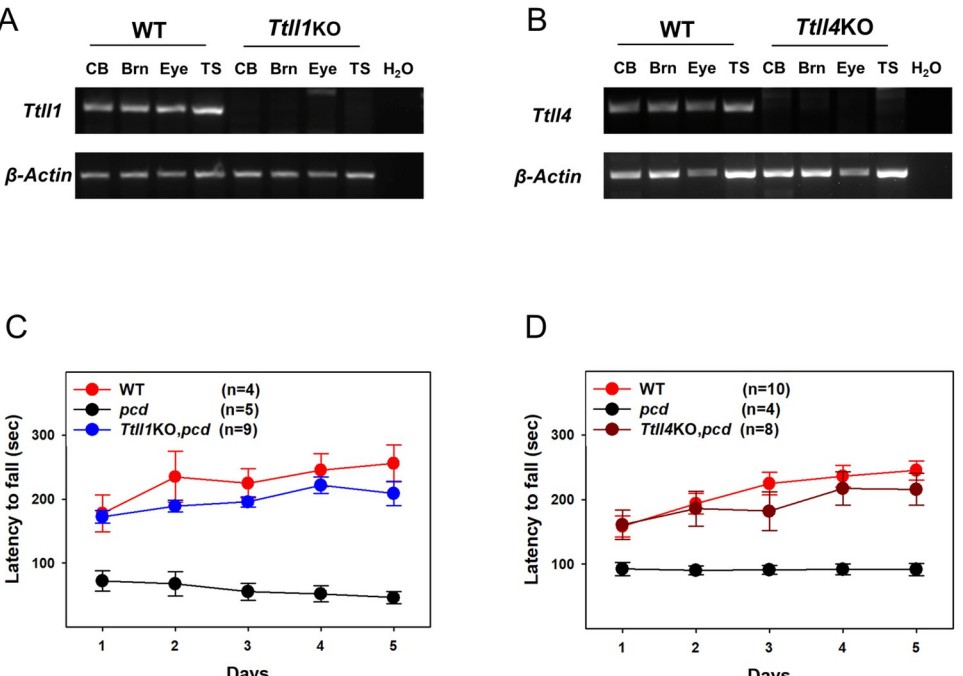

**Fig 2. TTLL1 and TTLL4 are required for Purkinje cell degeneration in *pcd* mice.** (A and B) RT-PCR using primers targeting deleted region in *Ttll1*KO (A) or *Ttll4*KO (B) allele confirmed the absence of *Ttll1*or *Ttll4* transcripts in cerebellum, brain, eye, and testis of respective KO mice. (C and D) Gender balanced littermates of each genotype (n = 4–10/genotype) at 7 weeks of age were tested on a standardized accelerating rota-rod for five consecutive days. The latency to fall in seconds (mean ± SEM) was not significantly different between WT and *Ttll1*KO,*pcd* (C) or *Ttll4*KO,*pcd* (D) groups (one-way ANOVA $p$ >0.05), whereas *pcd* mice were markedly impaired (C and D) (p < 0.05).

*Ttll1*KO,*pcd* (Fig 3C) or *Ttll4*KO,*pcd* mice (Fig 3D). In addition, there is widespread loss of Purkinje cells in 2-month-old *pcd* mice (Fig 3B and 3B'). In contrast, many calbindin-positive Purkinje cells are evident throughout all cerebellar lobes examined in *Ttll1*KO,*pcd* (Fig 3C and 3C') and *Ttll4*KO,*pcd* mice (Fig 3D and 3D'). Higher power images (Fig 3A'–3D') confirm the presence or absence of Purkinje cells and indicate that the morphology of surviving Purkinje cells does not vary noticeably between wild-type mice and *Ttll1*KO,*pcd* or *Ttll4*KO,*pcd* mice. The cerebella of *Ttll5*KO,*pcd*, *Ttll7*KO,*pcd* and *Ttll11*KO,*pcd* were also small compared to wild-type mice (S3A–S3D Fig). In addition, deletion of TTLL5, 7 or 11 had no impact on Purkinje cell degeneration in *pcd* (S3B'–S3D' Fig).

That loss of TTLL1 rescues Purkinje cell death in *pcd* was known [7,22], but loss of TTLL4 protecting these cells is a novel finding. As Purkinje cell loss is progressive in *pcd*, we considered the possibility that TTLL4 loss might slow but not prevent their death. Therefore, we compared 10-month-old wild type and *Ttll4*KO,*pcd* mice (Fig 4A). The size of cerebellum in *Ttll4*KO,*pcd* mice (Fig 4Ab) is not different from wild-type of the same age (Fig 4Aa). Again, many calbindin-positive Purkinje cells were observed in all lobes examined (Fig 4Ab and 4Ad), indicating that the rescue is relatively long term.

We also wanted to compare the degree of Purkinje cell protection conferred by TTLL1- or TTLL4-deficiency. Anatomical reconstruction of the cerebellum would be a daunting undertaking, so we sought a quantitative orthologous method. Previously, we identified many markers of Purkinje cells using molecular profiling [29]. Therefore, we examined RNA expression of three such markers [30–32] as a surrogate for Purkinje cell numbers in 2 months old mice. Three marker genes were chosen to minimize the possibility that expression of a single marker

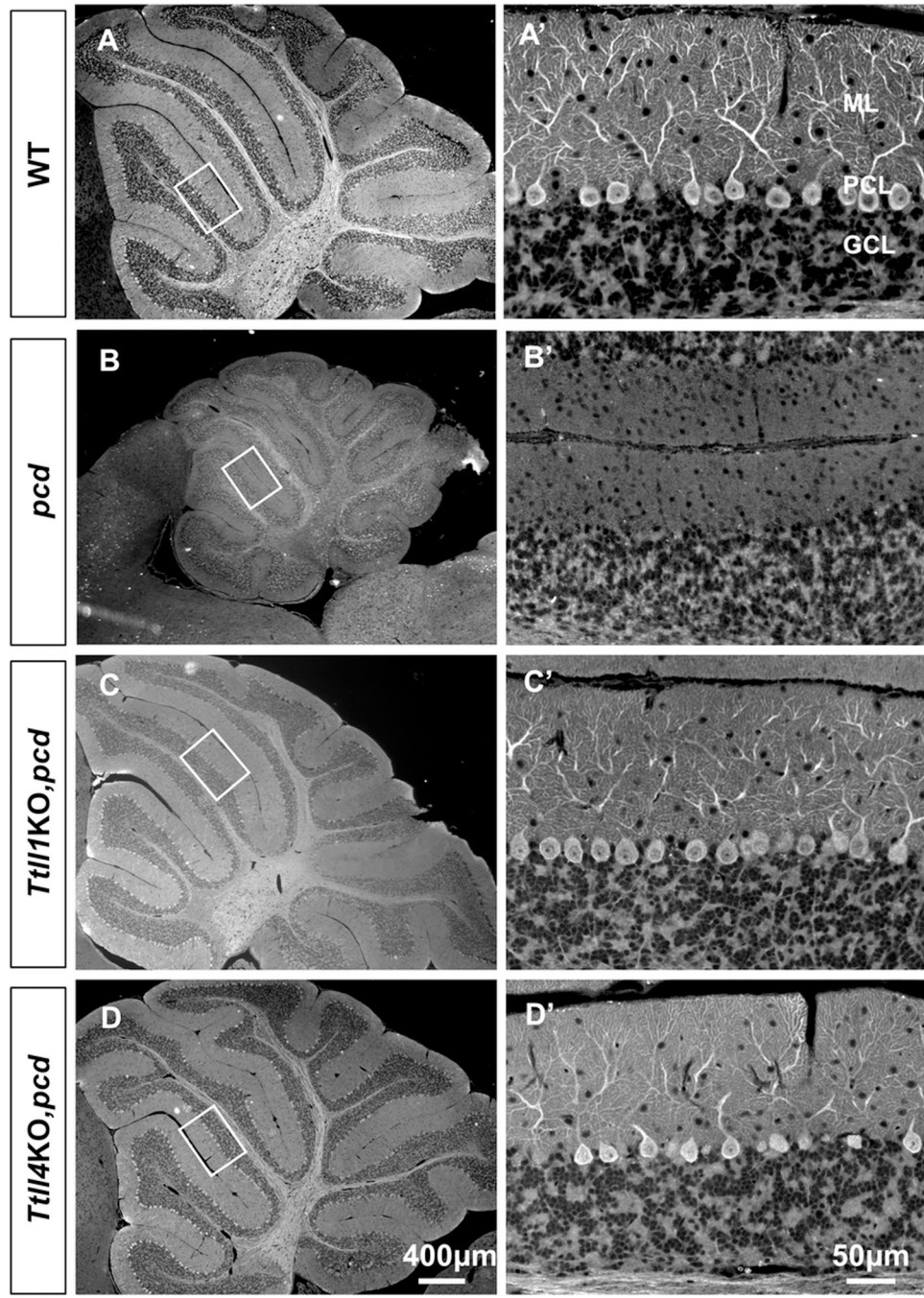

**Fig 3. Loss of TTLL1 and TTLL4 attenuated Purkinje cell degeneration in *pcd* mice.** (A-D) Calbindin D-28K immunofluorescence staining of cerebellar sections from 2-month old wild-type (A and A'), *pcd* (B and B'), *Ttll1*KO, *pcd* (C and C') and *Ttll4*KO,*pcd* (D and D'). Note the cerebellum of *pcd* mice (B) is smaller than that of wild-type (A), whereas *Ttll1*KO or *Ttll4*KO restored the size of the cerebellum in *pcd* mice. (A'-D') 20X view of boxed areas in A-D, respectively showed that calbindin-positive Purkinje neurons are lost in *pcd* mice (B'), whereas *Ttll1*KO,*pcd* (C') or *Ttll4*KO,*pcd* (D') did not exhibit overt Purkinje cell degeneration. ML: Molecular Layer; PCL: Purkinje Cell Layer; GCL: Granule Cell Layer.

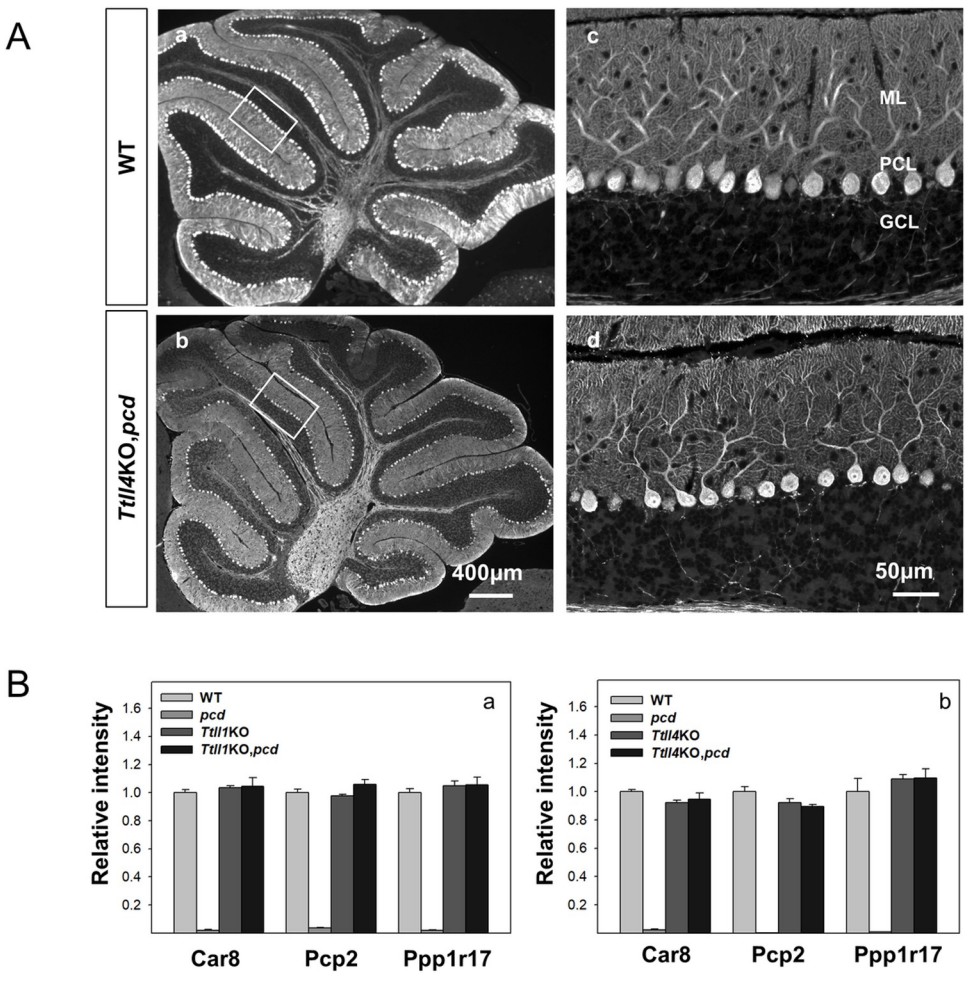

**Fig 4. Loss of TTLL4 attenuates Purkinje cell degeneration in older *pcd* mice.** (A) Calbindin D-28K immunofluorescence staining of cerebellar sections from 10-month old wild-type (a and c) and *Ttll4*KO,*pcd* (b and d) mice. Note the size of cerebellum of *Ttll4*KO,*pcd* is similar to that of wild-type mice and many Purkinje cells survive in *Ttll4*KO,*pcd* double mutant (d). (B) Quantitative RNA expression analysis of three Purkinje cell markers, Car8, Pcp2, and Ppp1r17 in cerebella from *Ttll1*KO (a) or *Ttll4*KO (b) genotypes. For comparison purposes, data (mean ±SEM) have been normalized to values in wild-type mice Three independent mice per genotype. Expression of the 3 markers is low or undetectable in *pcd* cerebellum (a and b) reflecting the massive loss of Purkinje cells. In contrast, there are no statistical differences between the expression of the markers in wild-type (WT); *Ttll1*KO; *Ttll4*KO; *Ttll1*KO,*pcd* and *Ttll4*KO,*pcd* mice.

might be directly affected by glutamylation. Expression of all three markers was almost undetectable in *pcd* cerebellum (Fig 4B), reflecting the massive loss of Purkinje cells. In contrast expression of the three markers was statistically the same in wild-type (Fig 4Ba and 4Bb), *Ttll1*KO,*pcd* (Fig 4Ba) and *Ttll4*KO,*pcd* cerebellum (Fig 4Bb). This indicates that the degree of Purkinje cell rescue by loss of *Ttll1* in *pcd* is indistinguishable from that conferred by loss of *Ttll4* at this age.

## Loss of TTLL1 or TTLL4 variably affect tubulin polyglutamylation in *pcd* cerebellum

In *pcd*, tubulin is hyperglutamylated [2] and loss of TTLL1 is reported to reduce glutamylation levels in cerebellum in parallel with its ability to rescue Purkinje cell degeneration [7].

Therefore, we compared tubulin glutamylation levels in cerebellum by immunoblotting of cerebellar extracts using the GT335 (detects gamma-linked branching glutamate [33]) and polyE (detects 3 or more consecutive glutamate residues residing at the C-terminus of a chain [1,6,34]) antibodies. In agreement with previous studies [2,7], tubulin glutamylation is increased in *pcd* mouse cerebellum and is markedly decreased in *Ttll1*KO and *Ttll1*KO,*pcd* cerebellum (Fig 5A). In contrast, loss of TTLL4 has no marked effect on tubulin polyglutamylation either alone or in *pcd* mice (Fig 5B). We also quantified the levels of glutamylation and confirm the reduction in *Ttll1*KO animals and the lack of statistically significant difference between *pcd* and *Ttll4*KO,*pcd* mice (Fig 5A and 5B). Nevertheless, the current analysis does demonstrate that TTLL4 loss of function attenuates Purkinje cell loss up to at least 10-months of age. In addition, we show that although loss of TTLL7 does not spare Purkinje cells in *pcd* mice, it does reduce polyglutamylation signal (S2C Fig). During revision of this manuscript a study appeared that also demonstrated reduced tubulin polyglutamylation but failure to rescue Purkinje cells in *Ttll7*KO,*pcd* mice [35]. These authors also showed that while TTLL1 is considered an α-tubulin preferring enzyme, it can glutamylate β-tubulin in the presence of TTLL7 [35].

A caveat with immunoblotting is that it is a bulk assay in which Purkinje cells are a relatively minor component and it is possible that there are changes in glutamylation in Purkinje cells that are masked by signal from other cellular components. To address this, we performed immunostaining of cerebellum with the GT335 antibody. As Purkinje cells degenerate in adult *pcd*, we first examined GT335 staining in postnatal day 19 (P19) mice, prior to Purkinje cell loss. In wild type mice Purkinje cells had prominent GT335 staining, whereas the granule cell layer was only weakly stained (Fig 6A). In *pcd* and *Ttll4*KO,*pcd* mice, there was no obvious difference in the GT335 staining of Purkinje cells compared to wild-type although the granule cell layer was more prominently stained in both (Fig 6B and 6C). To establish specificity of staining we absorbed the antibody with porcine brain tubulin and saw a marked reduction in immunofluorescence (S4 Fig).

In adult wild-type mice, Purkinje cells have prominent GT335-immunoreactivity particularly in the cell soma and principle dendrites (Fig 6D). In contrast, there are low levels of GT335 immunoreactivity in the granule cell layer. In adult *pcd* mice, Purkinje cells are absent but the granule cell layer is more intensely stained than in wild-type mice (Fig 6E). In adult *Ttll4*KO,*pcd* mice, Purkinje cells are preserved and are GT335-immunoreactive (Fig 6F), although they are not noticeably different from the wild-type. However, the granule cell layer is still intensely GT335-positive (Fig 6F). This suggests the increase in GT355 immunoreactivity on western blots (most likely α-tubulin, based on comparison of its migration with other immunoblotting studies in *pcd* [2]), represents the contribution from granule neurons. Furthermore, our data indicate that *Ttll4* loss does not affect the glutamylation of α-tubulin in granule neurons. In sum, loss of Nna1/CCP1 elicits increased tubulin glutamylation in granule neurons and is independent of TTLL4. However, there are no marked changes in GT335 immunoreactivity in Purkinje cells.

## TTLL1 and TTLL4 also contribute to degeneration of olfactory bulb mitral cells in *pcd* mice

Other neuronal populations degenerate in *pcd* mice [17,25] and we next determined whether other TTLLs and especially TTLL1 or TTLL4 play any role in this process. Mitral cells of the olfactory bulb degenerate in *pcd* mice [36] albeit on a slower time course than Purkinje cells [17]. Mitral cells are output neurons central to odorant perception whose dendrites form synapses with olfactory nerve axons in structures called glomeruli and whose axons project via the

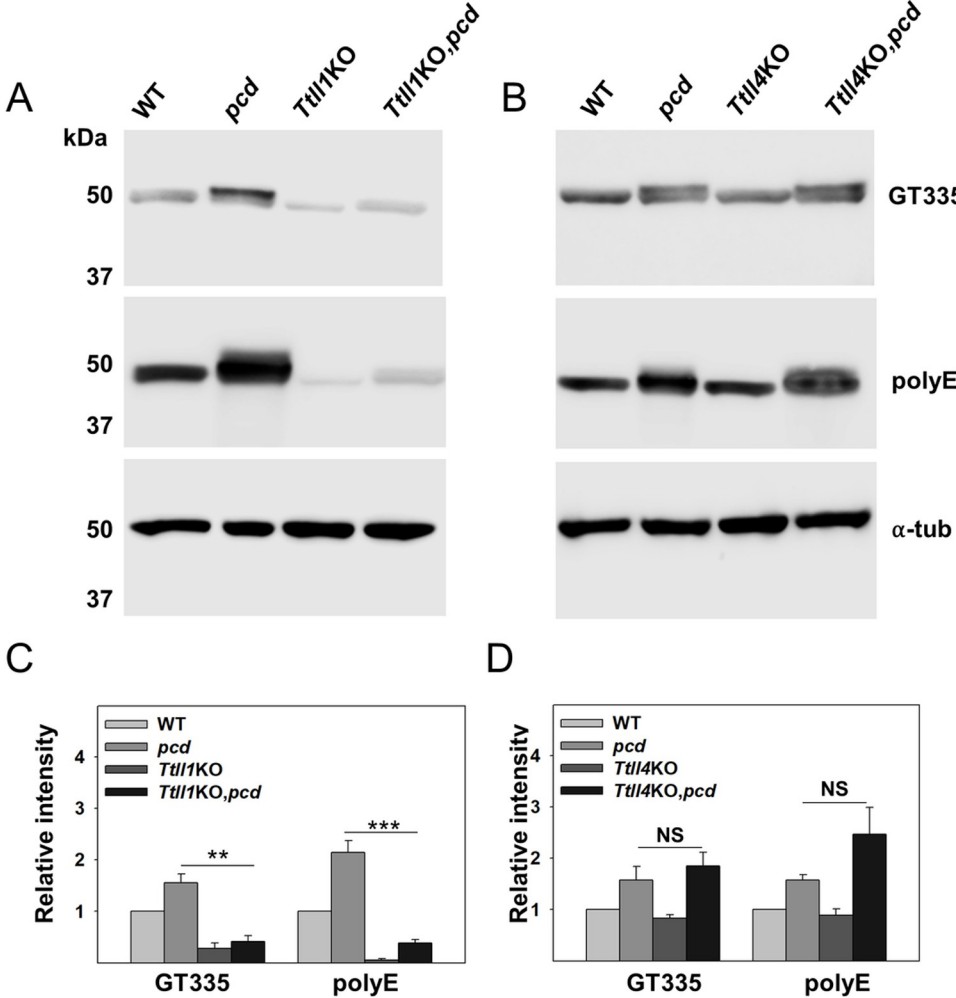

**Fig 5. Loss of TTLL1 or TTLL4 differentially affects tubulin polyglutamylation in wild-type and *pcd* cerebellum.**
(A) Representative western blot of polyglutamylated tubulins in cerebellar lysates from wild-type (WT), *pcd*, *Ttll1*KO, and *Ttll1*KO,*pcd* mice detected using GT335 and polyE antibodies. In *pcd*, there is increased immunoreactivity to both GT335 and polyE antibodies compared to wild-type. However, GT335- and polyE- immunoreactive bands were greatly reduced in *Ttll1*KO and *Ttll1*KO,*pcd*. (B) In contrast to *Ttll1*KO, loss of *Ttll4* function did not alter the basal or elevated polyglutamylation levels in *pcd*. (C and D) Quantitative analysis of intensity of GT335 or polyE immunoreactive bands using LiCOR with normalization to α-tubulin levels. Whereas *Ttll1*KO caused a significant reduction in GT335 and polyE signals, loss of *Ttll4* had no significant affect. The bars represent the mean ±SEM of 4–6 animals of each genotype. ** $p < 0.01$; *** $p < 0.001$; NS: Not significant.

lateral olfactory tract to the olfactory cortex [37]. We assessed mitral cell survival using Tbr2 immunostaining as a marker (Fig 7) [38]. In 5-month-old wild type mice, Tbr2-positive cells are present in two layers (Fig 7A, 7A' and 7E). Mitral cells are aligned in a layer (mitral cell layer, MCL) at the junction of the granule cell layer (GCL) and external plexiform layer (EPL) (Fig 7A, 7A' and arrows in 7E). Tbr2-positive cells (predominantly tufted cells) are also evident in a zone encompassing the boundary of the glomerular (GL) and external plexiform layers (Fig 7A–7D and 7A'–7D'). In *pcd* mice the olfactory bulb is shrunken and Tbr2-positive cells in the mitral cell layer are markedly reduced (Fig 7B, 7B' and 7F). In contrast, there is no overt impact on other Tbr2-positive neurons (Fig 7B and 7B') consistent with earlier studies showing survival of tufted cells in *pcd*[36]. In both *Ttll1*KO,*pcd* (Fig 7C, 7C' and 7G) and *Ttll4*KO,

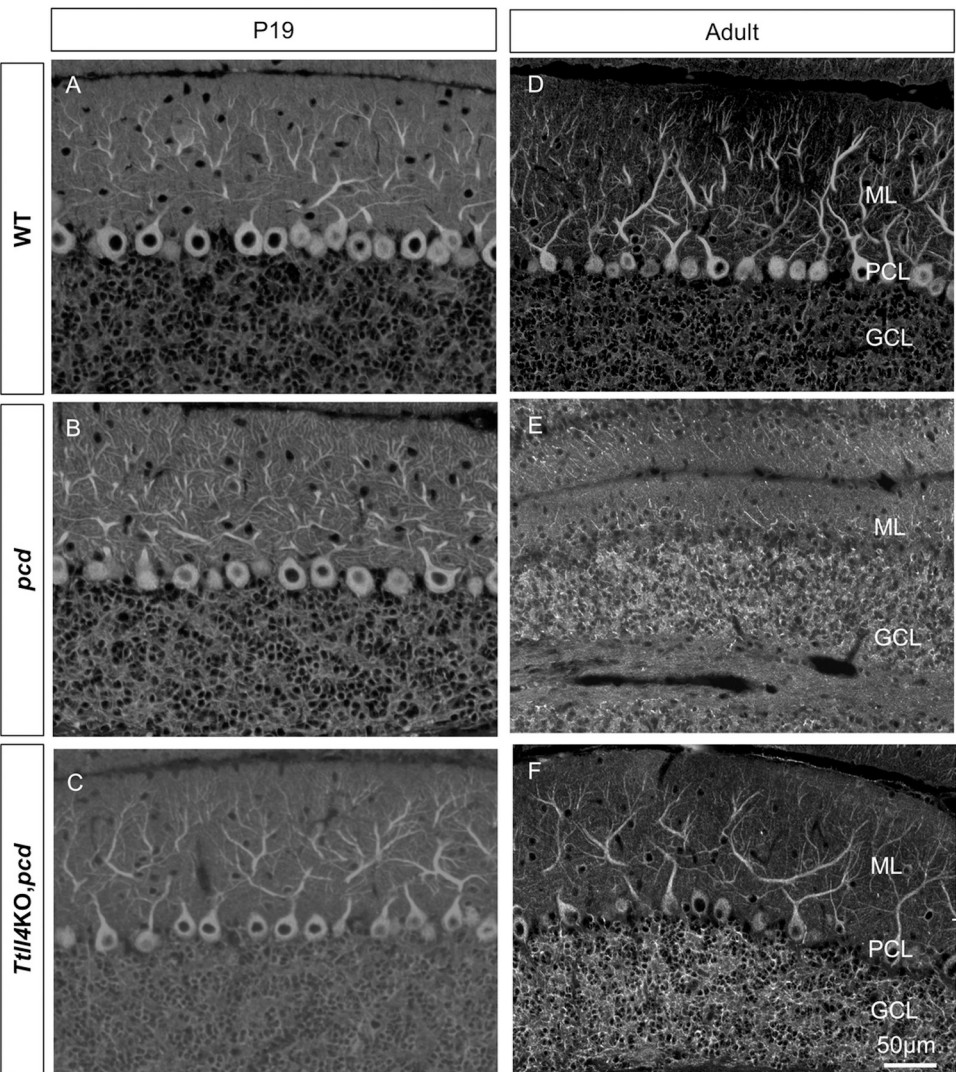

**Fig 6. Increased GT335 signal in granule cells of *pcd* mice is not affected by *Ttll4* loss of function.** Cerebellar sections from 19 days old (A-C) or adult (D-F) wild-type (A and D), *pcd* (B and E) or *Ttll4*KO,*pcd* (C and F) mice stained with GT335 antibody. (A-C) At P19 prior to Purkinje cell degeneration, GT335-immunoreactivity is prominent in the Purkinje cell soma and principal dendrites of all three genotypes. However, there is no apparent difference in Purkinje cell GT335 immunoreactivity in *Ttll4*KO,*pcd* (C) compared with wild-type (A) or *pcd* (B). In contrast, there appears to be increased GT335 activity in the granule cell layer of *pcd* and *Ttll4*KO,*pcd* mice. (D-F) In adult cerebella, the GT335 immunoreactivity in granule cells remains higher in *pcd* (E) and *Ttll4*KO,*pcd* (F) compared to wild-type (D). As in young animals, there were no obvious differences in GT335 expression pattern in Purkinje cells of wild-type (D) versus *Ttll4*KO,*pcd* mice (F). ML: Molecular Layer; PCL: Purkinje Cell Layer; GCL: Granule Cell Layer.

*pcd* (Fig 7D, 7D' and 7H) mice, there are more Tbr2-positive cells in the MCL compared with *pcd* mice. There was no overt increased survival of mitral cells in the other *Ttll*KO strains on a *pcd* background (S5 Fig).

To determine whether loss of TTLL4 function slows rather than prevents mitral cell degeneration, we assessed survival in 7- and 10-month-old wild-type and *Ttll4*KO,*pcd* mice. As in younger animals, two bands of Tbr2-positive cells are present throughout the olfactory bulbs of both wild-type and *Ttll4*KO,*pcd* mice (Fig 8A and 8B), and many Tbr2-positive cells are present in the MCL of both genotypes at 7 and 10 months (Fig 8). Therefore, as with Purkinje

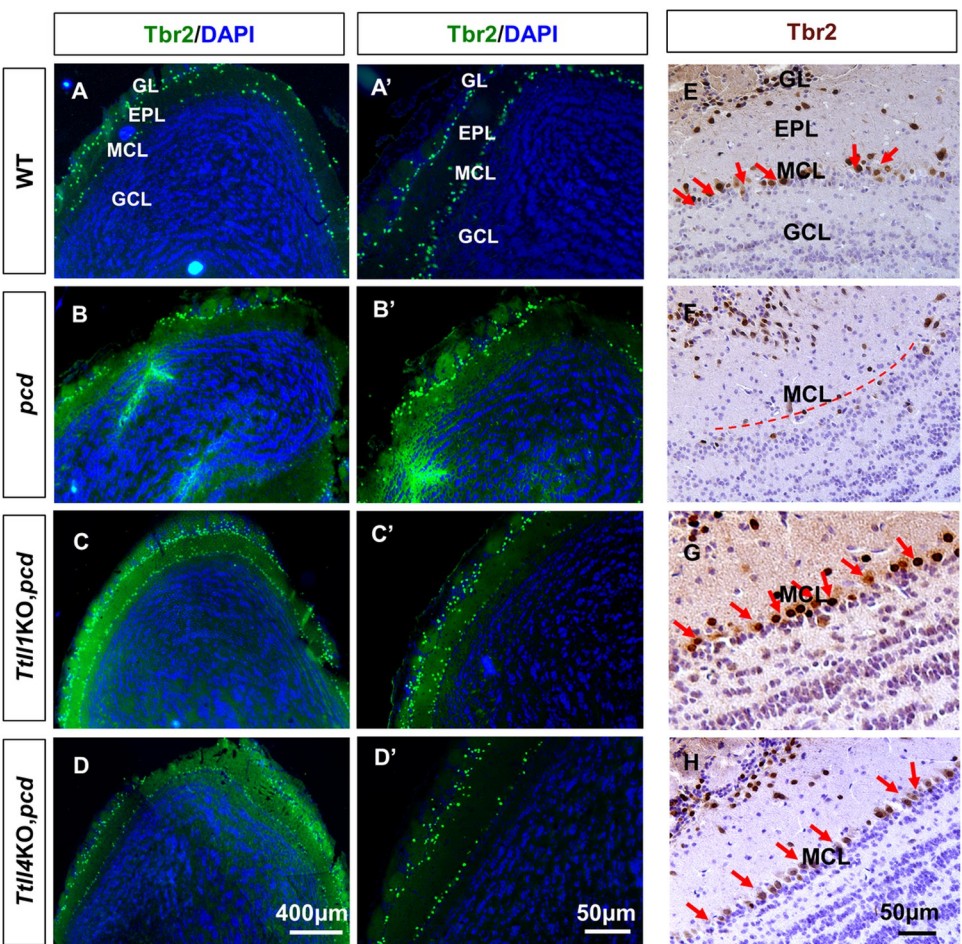

**Fig 7. *Ttll1* and *Ttll4* are required for olfactory bulb mitral cell degeneration in *pcd* mice.** Sections of olfactory bulbs from 5-month-old wild-type (A-A', E), *pcd* (B, B', and F), *Ttll1*KO,*pcd* (C,C', and G), and *Ttll4*KO,*pcd* (D, D', and H) were immunostained with anti-Tbr2 which recognizes mitral cells and tufted cells. (A-D) low power immunofluorescence images of Tbr2 expression with DAPI counter staining. (A'-D') Higher magnification images shown in A-D, respectively. Tbr2-positive cells are present in 2 distinct layers. The outer layer is comprised predominantly of tufted cells that are preserved in all genotypes. In contrast the inner layer is predominantly composed of olfactory bulb mitral cells are largely lost in *pcd* mice at this age (B, B'), whereas concomitant elimination of either *Ttll1* (C and C') or *Ttll4* (D and D') in *pcd* mice showed more Tbr2-positive mitral cells. (E-H) Bright-field images of IHC also showed that Tbr2-positive mitral cells are lost in *pcd* mice (compare E to F). Red arrows indicate Tbr2-positive mitral cells in wild type mice and dashed red lines denotes the location of the mitral cell layer in *pcd*. Note that in *Ttll1*KO,*pcd* (G) and *Ttll4*KO,*pcd* (H) double mutants more Tbr2-positive mitral cells were evident (red arrows). GL: Glomerular Layer; EPL: External Plexiform Layer; MCL: Mitral Cell Layer; GCL: Granule Cell Layer.

cells in cerebellum, loss of TTLL4 provides a sustained attenuation of olfactory bulb mitral cells loss in *pcd* mice.

## TTLL1 and TTLL4 differentially contribute to photoreceptor degeneration in *pcd* mice

We next examined whether knocking out any *Ttll* attenuated photoreceptor degeneration in *pcd* mice. Retinae of 5-month-old mice were immunostained for the rod-specific rhodopsin, and nuclei visualized by DAPI. In wild-type mice, the outer nuclear layer (ONL), which comprises the cell bodies of photoreceptors, is 10–12 nuclei deep (~51 μm) and rhodopsin

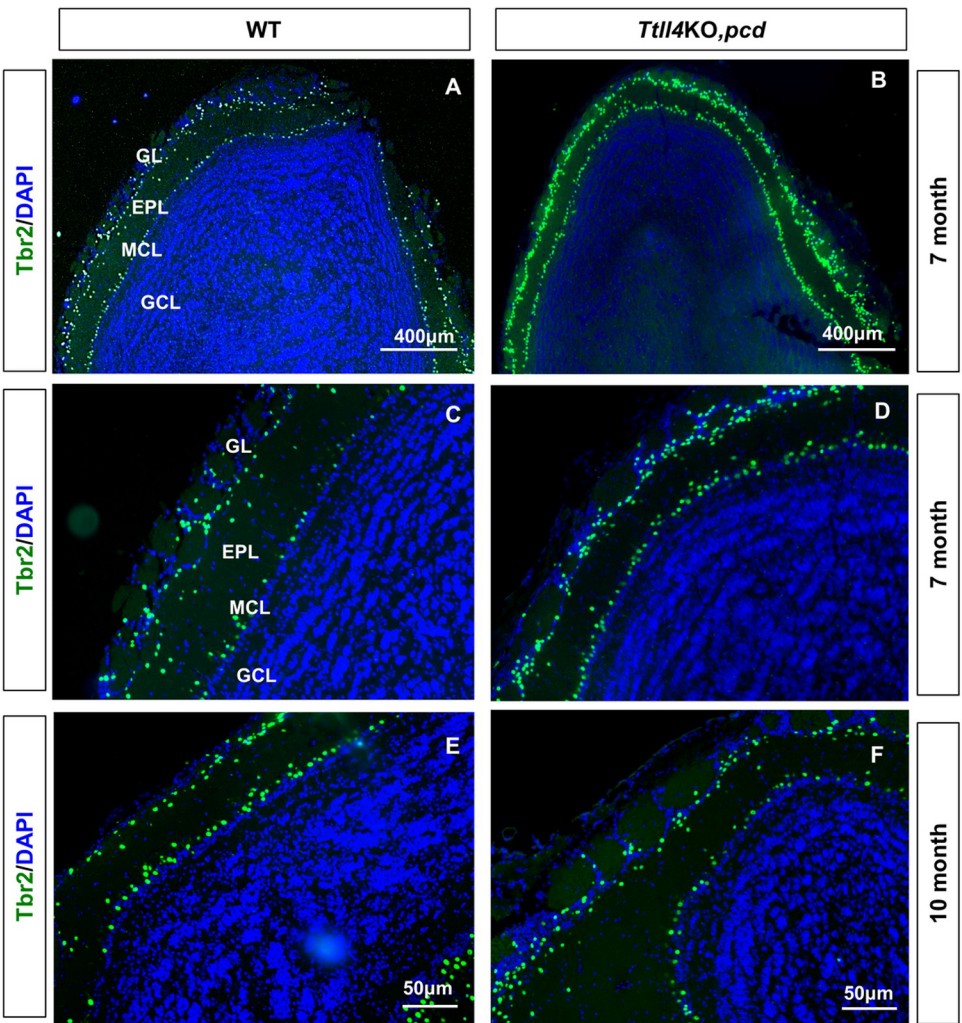

**Fig 8. Survival of mitral cells in older *Ttll4*KO,*pcd* mice.** Sections of olfactory bulbs from 7-month-old wild-type (A and C), *Ttll4*KO,*pcd* (B and D) and, 10-month-old wild-type (E) and *Ttll4*KO,*pcd* (F) mice were immunostained with anti-Tbr2 and counter stained with DAPI. Expression of Tbr2 is prominent in two layers throughout olfactory bulb of wild-type (A and C) and *Ttll4*KO,*pcd* (B and D) mice at 7-month of age. The same pattern of Tbr2 was evident in olfactory bulb of wild-type (E) and *Ttll4*KO,*pcd* (F) at 10-month. GL: Glomerular Layer; EPL: External Plexiform Layer; MCL: Mitral Cell Layer; GCL: Granule Cell Layer.

immunoreactivity is restricted to the outer segment layer (OSL) (Fig 9Aa and 9Ae). In *pcd*, the ONL is much thinner (~ 15 μm) comprising 2–3 nuclei (Fig 9Ab), reflecting substantial photo-receptor loss. In addition, in *pcd* retinae the OSL is thinner and rhodopsin immunoreactivity is observed not only in the OSL (Fig 9Ab) but also in the ONL (Fig 9Af). The mis-location of rhodopsin is considered a pathological finding indicative of impaired trafficking of proteins from the photoreceptor cell body to the outer segment via the specialized connecting cilium [39,40]. In *Ttll1*KO,*pcd* mice, the thickness of the ONL is similar to wild-type values (~10–12 nuclei, ~45 μm), and the length of the OSL is also comparable to that of wild-type (Fig 9Ac) and there is no mis-location of rhodopsin in the cell body (Fig 9Ag). Therefore, at 5-month of age the photoreceptor degeneration in *pcd* mice is largely prevented by *Ttll1* loss-of-function. In the retinae of *Ttll4*KO,*pcd* double mutants, the ONL was about 5–7 nuclei in depth (~31μm), which is more than that of *pcd* but less than wild-type (Fig 9Ad). The OSL in

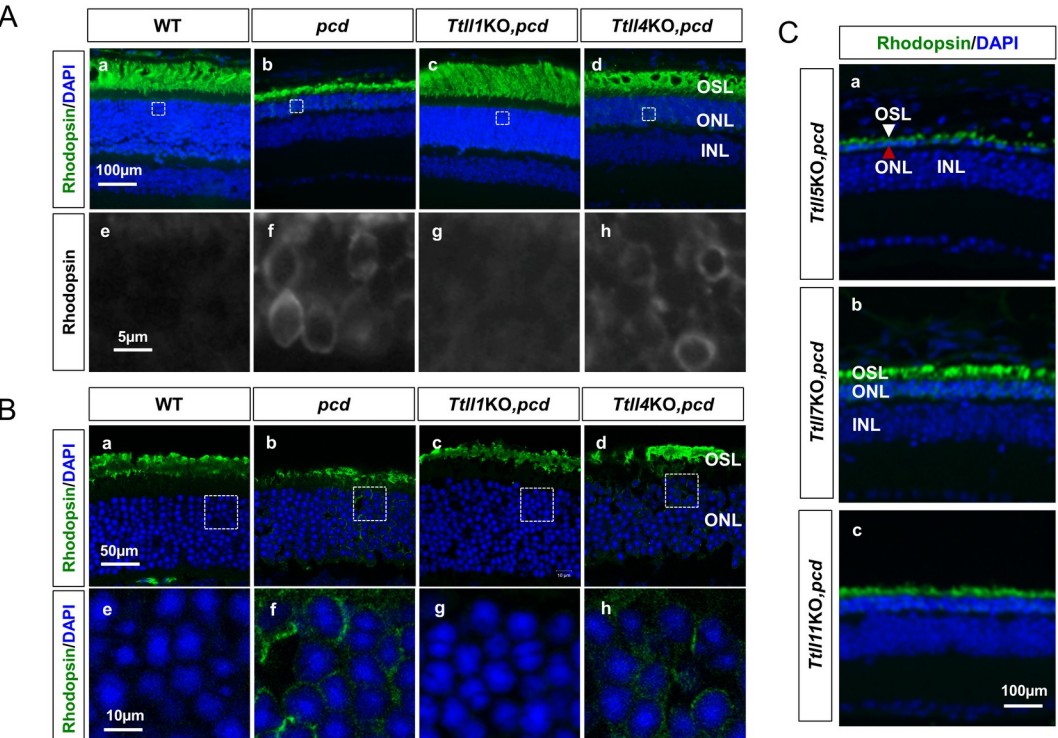

**Fig 9. TTLLs contribute to photoreceptor degeneration in *pcd* mice.** (A) Sections of retinae from 5-month-old wild-type (a), *pcd* (b), *Ttll1*KO,*pcd* (c), and *Ttll4*KO,*pcd* (d) mice were immunostained with the rod photoreceptor marker, rhodopsin (green) and nuclei visualized with DAPI (blue). (e-h) Higher power view of rhodopsin staining in the boxed areas in a-d. (a) The outer nuclear layer (ONL) of wild-type retina is about 10–12 nuclei thick, and rhodopsin is exclusively localized in the outer segment layer (OSL) and absent in ONL (e). (b) The ONL of *pcd* retina is reduced to about 2–3 nuclei in depth and is accompanied by the mis-localization of rhodopsin to cell bodies (f). (c) The number of nuclei in the ONL of *Ttll1*KO,*pcd* retina is similar to wild-type and there is no mis-localization of rhodopsin (g). (d) In *Ttll4*KO,*pcd* retina, the number of nuclei in the ONL were greater than in *pcd* (b), but less than wild-type (a) or *Ttll1*KO,*pcd* (c) mice. The mis-localization of rhodopsin was still observed in *Ttll4*KO,*pcd* retina (h). (B) Sections of retinae from P30 wild-type (a), *pcd* (b), *Ttll1*KO,*pcd* (c), and *Ttll4*KO,*pcd* (d) immunostained with rhodopsin (green) and DAPI (blue). (e-h) Higher power image of boxed areas in a-d respectively. In wild-type retina, rhodopsin is exclusively localized in the OSL (a and e). The number of nuclei in the ONL in *pcd* (b) is comparable to wild-type mice (a), but rhodopsin mis-location to the cell body is already evident (f). In *Ttll1*KO,*pcd* retina, there is no rhodopsin mis-location (g). In *Ttll4*KO,*pcd* retinae, rhodopsin is mis-located in the photoreceptor cell body (h). (C) Sections of retina from 5-month *Ttll5*KO,*pcd* (a), *Ttll7*KO,*pcd* (b) and *Ttll11*KO,*pcd* (c) were stained as in (A). Loss of function mutants for these *Ttll*s do not increase the survival of the photoreceptors on the *pcd* background (compare panel Ab with panels Ca-c). However, in *Ttll5*KO,*pcd* retina, there is a near ablation of the ONL (a) with only a single layer of cells containing many gaps. Red and white arrows indicate DAPI positive and rhodopsin positive cells in the ONL. OSL: Outer segment layer; ONL: Outer nuclear layer; INL: Inner nuclear layer.

*Ttll4*KO,*pcd* retina is also substantially broader than *pcd* (Fig 9Ad). However, some retention of rhodopsin in cell bodies was evident (Fig 9Ah). Therefore, *TTLL4*-deficiency partially attenuated or slowed photoreceptor degeneration in *pcd* mice.

Rhodopsin retention in the cell body is a sign of aberrant protein transport but may also be the secondary consequence of outer segment degeneration. Therefore, we examined the localization of rhodopsin in retina at age P30, when retina differentiation and maturation are just complete and prior to photoreceptor loss [17,41]. At P30, the depth of the ONL was comparable between the wild-type and *pcd* retinae (Fig 9Ba and 9Bb). However, rhodopsin was already evident in the cell body of *pcd* photoreceptors (Fig 9Bf). These results suggest that rhodopsin transport is directly impaired by Nna1 dysfunction in *pcd* mice. In *Ttll1*KO,*pcd* mice no mis-localized rhodopsin was evident (Fig 9Bg). On the other hand, mis-localized rhodopsin was

evident in *Ttll4*KO,*pcd* retina, although it was less prominent than in *pcd* mice (Fig 9Bh). Therefore, the aberrant localization of rhodopsin observed in *Ttll4*KO,*pcd* mice at 5-month of age is already evident at P30 and confirm eliminating *Ttll4* does not fully rescue the deficit in rhodopsin transport in *pcd* mice.

Besides TTLL1 and TTLL4 loss only TTLL5-deficiency had any influence on photoreceptor degeneration in *pcd* (Fig 9C). Mutations of TTLL5 have been linked to retinal disease in humans [42] and a *Ttll5* mutant mouse is reported to undergo late (~20 months-old) photoreceptor degeneration [43]. At 5-month of age, we did not observe overt loss of photoreceptors in *Ttll5*KO mice. However, *Ttll5*KO,*pcd* mice lose almost all photoreceptors and only a single layer of cell bodies survives in the ONL (Fig 9Ca). Thus, loss of anabolic TTLL5 synergizes with loss of catabolic Nna1 to promote photoreceptor degeneration.

## Neither *Ttll1*KO nor *Ttll4*KO rescues male infertility in *pcd* mice

The *pcd* male is infertile due to defects in spermatogenesis associated with testicular structural anomalies and cell death in the germinal epithelium [10,44]. TTLL1 (the gene targeted here) is the catalytic subunit of a multiprotein complex [26] and a mutation in another component of the complex (PGs1) behaves as a TTLL1-loss of function allele [24,45]. The PGs1mutant also exhibits male infertility and has testicular anomalies [45]. Progeny testing showed that the *Ttll1*KO males used here were also infertile. Cell death in the germinal epithelium was characterized by the presence of pyknotic and/or multinucleated giant cells, the latter being a specific form of germ cell degeneration caused by the fusion of spermatids (Fig 10, compare panel Aa with panels Ab, Ac, Dd and Ff; arrows indicate dying cells). Given that depletion of *Ttll1* or *Ttll4* ameliorated neuronal degeneration in *pcd* mice, we assessed whether knocking out *Ttll1* or *Ttll4* could also rescue male infertility.

Hematoxylin-eosin staining revealed a striking cell loss in the germinal epithelium of *Ttll1*KO testes (Fig 10 Ac and 10Ac's, arrows indicate dying cells). These morphologic changes were more severe than those reported for PGs1 mutants [45] and even those of *pcd* (Fig 10Ab and 10Ab'). The *Ttll1*KO,*pcd* testis, although still abnormal, had substantially improved overall morphology and reduced cell loss in the germinal epithelium (Fig 10Ad and 10Ad') compared to either *Ttll1*KO or *pcd*. As expected, the sperm count of *Ttll1*KO mice was significantly lower than that of the wild-type (Fig 10B). However, despite the improved testicular cytoarchitecture, the sperm count from *Ttll1*KO,*pcd* double mutants was not significantly different from that of either *Ttll1*KO or *pcd* (Fig 10B).

In contrast to *Ttll1* mutants, progeny testing showed that male (and female) *Ttll4*KO mice were fertile. Indeed, the morphology of the testes and sperm counts from *Ttll4*KO mice were comparable with that of wild-type littermates (Fig 10Ae, 10Ae', 10Aa, 10Aa' and 10B). The testicular structure in *Ttll4*KO,*pcd* double mutants (Fig 10Af and 10Af') was improved compared with *pcd* mice, but substantial numbers of dead cells were still evident (Fig 10Af' arrows). Despite the improved testicular morphology, the sperm count from *Ttll4*KO,*pcd* was still as low as that of *pcd* mice (Fig 10B).

## Discussion

A growing body of work has demonstrated that appropriately regulated polyglutamylation is required for neuronal survival across species [2,18,19] and also contributes to axonal regeneration [15,46] and spermatogenesis [10,27,45,47]. In this study we investigated the contribution of anabolic TTLL enzymes to polyglutamate homeostasis and degenerative phenotypes in *pcd* mice that are deficient in the catabolic Nna1/CCP1 enzyme [2,16]. We show that TTLL1 and TTLL4 are unique among the polyglutamylases as their loss counteracts to varying degrees the

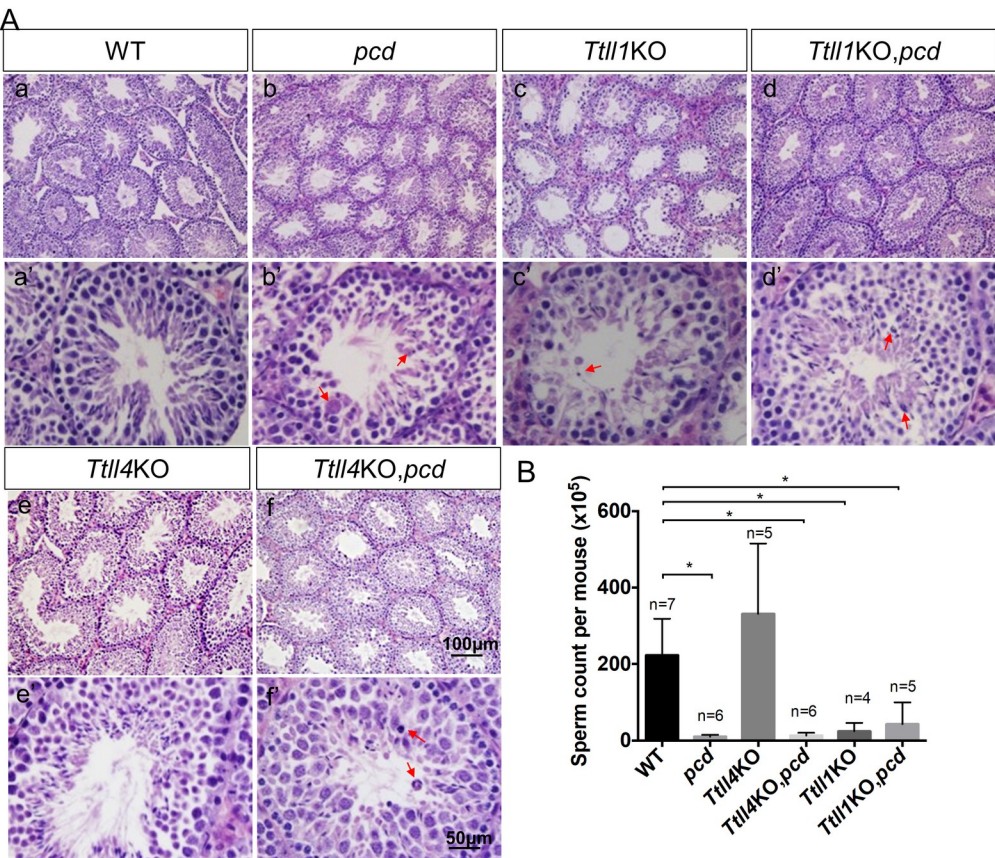

**Fig 10. Neither *Ttll1* nor *Ttll4* knockouts rescued male infertility of *pcd* mice.** (A) Hematoxylin and eosin staining on sections of testis from 3-month-old wild-type (a and a'), *pcd* (b and b'), *Ttll1*KO (c and c'), *Ttll1*KO,*pcd* (d and d'), *Ttll4*KO (e and e'), and *Ttll4*KO,*pcd* (f and f') mice. Compared to wild-type (a and a') mice, the elongating spermatids were reduced in *pcd* testis (b and b'). Knocking out *Ttll1* led to destruction of the germinal epithelium and cell loss in testis (c and c'), and partially restored testicular morphology in *pcd* mice (d and d'), although marked cell loss was still evident (arrows). *Ttll4*KO did not alter the general morphology of testis compared to that of wild-type testis (e and e'), and had a marginal effect in increasing the thickness of the germinal epithelium and the diameter of seminiferous tubules in *pcd* (f and f') but cell death was still evident (arrows). (B) Sperm counts from epididymis and vas deferens of wild-type; *pcd*; *Ttll4*KO; *Ttll4*KO,*pcd*; *Ttll1*KO and *Ttll1*KO,*pcd* mice. *Ttll4*KO mice are fertile, and their sperm counts are indistinguishable from wildtype ($p = 0.19$) (Student's t test). In contrast, *Ttll1*KO and *pcd* mice are infertile and both have low sperm counts compared to wild type ($p = 0.002$, in both cases). Elimination of *Ttll1* or *Ttll4* failed to increase the sperm counts of *pcd* mice (*Ttll1*KO,*pcd*, $p = 0.25$; *Ttll4*KO,*pcd*, $p = 0.20$).

degenerative phenotypes in cerebellum, olfactory bulb and retinae of *pcd* mice (this study and [7,22]); providing evidence that the degeneration of different types of neurons in *pcd* mice likely share a common molecular pathogenesis involving hyperglutamylation.

An important question is the identity of the relevant substrate(s) for TTLL1 and TTLL4. The preference of TTLL1 for α-tubulin and TTLL4 for β-tubulin suggests that hyperglutamylation of both tubulin subunits is required for degeneration in *pcd*. However, this notion raises several questions. TTLL4 and TTLL7 are both expressed in Purkinje cells [5,28] and both initiate glutamylation of β-tubulin yet loss of TTLL7 does not counteract Purkinje cell degeneration in *pcd* (this study and [35]). Moreover, loss of TTLL7 reduces tubulin glutamylation in *pcd* (this study and [35]), whereas TTLL4 loss has no discernible effect either in bulk western blots or immunofluorescence analyses (Figs 5 and 6). This questions whether hyperglutamylation of β-tubulin contributes to the *pcd* phenotype and indeed whether tubulins are the relevant substrate for TTLL4 in this model. Non-tubulin substrates for polyglutamylation have

been identified [4,43,48] but whether any of these contribute to degeneration is unknown. A similar specificity paradox exists for TTLL1 and TTLL5 which are both expressed in Purkinje cells [28] and prefer α-tubulin [1,26], yet only TTLL1 loss prevents degeneration (Fig 3). There are caveats for interpreting these data. First, substrate specificities and catalytic properties of TTLLs should be more accurately viewed as preferences that were established largely *in vitro* [1,26] and with much less being known about their selectivity *in vivo* in specific cell types. Indeed, a recent study found that TTLL1, which is generally considered an elongator for α-tubulin, may function as an elongator on beta-tubulin *in vivo* [22,35]. Second, as proposed in the tubulin code, the various tubulin isoforms may be glutamylated at multiple sites [49,50] which could have distinct functional outcomes depending upon the pattern of residues glutamylated the lengths of the chains and the status of other tubulin post-translational modifications such as glycylation [51]. A further possibility is that critical glutamylation changes are confined to specific regions of the cell. To assess these parameters *in vivo* with single cell and sub-cellular specificity is challenging with the reagents and technologies currently available. Therefore, the conclusions that hyperglutamylation of β-tubulin is pathogenic and that it is the critical target substrate for TTLL4 *in vivo* requires further investigation.

If there is a common underlying pathogenic mechanism involving hyperglutamylation several observations remain to be explained. First, Nna1/CCP1 is broadly expressed in CNS neurons [16] and its elimination in *pcd* results in hyperglutamylation of tubulin throughout brain, yet only specific subsets of neurons degenerate [17]. Second, the tempo of degeneration of different cell classes varies, with some dying early, such as Purkinje cells whereas others, such as olfactory bulb mitral cells degenerate on an intermediate time scale and yet others, such as retinal photoreceptors, die progressively throughout a more protracted timeline [17,25]. How and whether these properties reflect differences in the balance of polyglutamate homeostasis (e.g. levels of various TTLLs and CCPs) or levels and subcellular distributions of different tubulin isoforms or other substrates in various cell classes remains to be fully established. Our systematic genetic analysis of TTLL dependencies in *pcd* mice provides perspectives on these issues.

Purkinje cells and mitral cells have relatively high levels of both TTLL1 and TTLL4 and both TTLLs contribute to degeneration in *pcd*. Therefore, intrinsic differences in the expression of TTLL1 and TTLL4 in distinct neuronal populations may be one determinant of their relative susceptibility to damage in response to perturbations that promote hyperglutamylation. Such intrinsic differences in TTLL1 and TTLL4 expression could also underlie the tempo of neuronal degeneration. Experiments are underway to test this hypothesis.

The degeneration of retinal photoreceptors highlights additional characteristics of the involvement of polyglutamylation in this process. Retinal photoreceptors appear particularly sensitive to changes in glutamylation status as their degeneration occurs not only in *pcd*, but also in mice or humans deficient in CCP5 [52,53] or TTLL5 [42,43] whose losses have no overt effects on other neurons [43]. In this instance sensitivity may relate to the specialized cellular architecture of photoreceptors. The rod outer segment of photoreceptors has no intrinsic translational capacity and proteins necessary for outer segment structure and function must be transported from the inner segment via a specialized connecting cilium [40]. The correct structure and function of the connecting cilium is essential for photoreceptor integrity and mutations in genes encoding its constituent proteins underlie several retinal diseases [39]. Furthermore, polyglutamylated tubulin is known to be associated with the axoneme of the connecting cilium [51] as well as cilia in general, centrioles, basal bodies, and flagella [54] and play a role in the proper formation of ciliated structures and trafficking along the cilia [55–57]. Indeed, hyperglutamylation of tubulin is associated with altered trafficking of vesicles and organelles in neurons of *pcd* mice although a causal relationship to degeneration has yet to be established [22]. The early-onset mis-localization of rhodopsin to the photoreceptor cell body

is regarded as an indicator of impaired trafficking through the connecting cilium [40] and is a prominent feature of the *pcd* retina both prior to and during receptor degeneration (Fig 9 and [51]). Our finding that TTLL1 deficiency not only attenuated photoreceptor loss but also eliminated the mis-localization of rhodopsin prior to receptor degeneration in *pcd* indicates it restores normal levels of transport through the connecting cilium and that impaired transport is the underlying cause of the degeneration. This notion is further supported by our finding that TTLL4-deficiency does not provide the same level of attenuation of photoreceptor loss as TTLL1 loss and only partially improves rhodopsin mis-localization (Fig 9). Given the ongoing impairment of rhodopsin trafficking in *pcd*,*Ttll4* KO mice it is probable that its loss of function slows rather than prevents photoreceptor degeneration in *pcd* mice. It also indicates that if TTLL1 and TTLL4 act in a common pathway the two are not equipotent, at least in the retina.

Mutations of the X-linked retinitis pigmentosa GTPase regulator (RPGR) cause photoreceptor degeneration [58]. Furthermore, a specific isoform of RPGR is a substrate for TTLL5 [43] and mutations in TTLL5 cause retinal dystrophy in humans [42,59] and slow photoreceptor degeneration in mice leading to the proposal that glutamylation of RPGR is required for its normal function [43]. We did not observe any noticeable photoreceptor loss in 5-month-old TTLL5-null mice, which is unsurprising as the reported loss was in mice of ~20-months of age [43]. However, 5-month-old *Ttll5*KO,*pcd* double mutants had a more severe photoreceptor loss compared to *pcd* alone (Fig 9C). Together this highlights two points: first, hyperglutamylation of one substrate can act cooperatively with hypoglutamylation of a different substrate to elicit photoreceptor degeneration; second, the tempo of degeneration is slower for hypoglutamylation compared to hyperglutamylation.

In testis, all enzymes involved in polyglutamylation are highly expressed [5] (Fig 1), potentially reflecting the importance of tubulin polyglutamylation in, for example, sperm flagellum formation. Despite their abundance, only the anabolic TTLL1, TTLL5, and TTLL9 [27,45,47], and the catabolic Nna1/CCP1 and CCP5[3,10,44] play key roles in spermatogenesis. In this study, *Ttll1* deficiency caused the most striking defects in testicular structure and morphology, worse even than *pcd* (Fig 10Ab and 10Ac). The abnormal morphology was accompanied by very low sperm counts (Fig 10B). In *Ttll1*KO,*pcd* double mutants, testicular morphology was improved compared to that of either single KO (compare Fig 10Ad with Fig 10Ab and 10Ac), however the marked deficit in sperm numbers was not rescued (Fig 10B). Therefore, TTLL1 and Nna1/CCP1 counteract each other in some but not all aspects of testicular function. In contrast to *Ttll*1KO mice, *Ttll*4KO males are fertile with no obvious testicular anomalies and normal sperm counts (Fig 10Ae and 10B). Nevertheless, knocking out *Ttll4* also rescued much of the aberrant testicular morphology in *pcd* (Fig 10Ab) but failed to rescue sperm counts (Fig 10B). This suggests that TTLL4 also generates substrates for Nna1/CCP1, but its function is dispensable in testis. Thus, the products of multiple TTLL polyglutamylases concomitantly contribute to male infertility in *pcd* mice.

Tubulin polyglutamylation levels in testis are generally low [10]. During male gametogenesis, exquisite regulation of tubulin posttranslational modification is required in multiple events such as spindle cleavage in mitosis and meiosis, sperm flagellum formation, trafficking in germ and Sertoli cells, and manchette formation and dismantlement [60]. It is conceivable that TTLL activity may be required in a transient manner during different events of spermatogenesis. This might be achieved through transcriptional regulation of their expression and potentially allosteric mechanisms as well as by the levels and activity of CCPs that metabolize their substrates. In contrast to markedly impaired male reproductive capacity, female reproduction is essentially normal in all *Ttll*KO strains examined. This is despite the fact that tubulin posttranslational modification plays a role during oocyte maturation [60] and motility of cilia in the Fallopian tube epithelium is required for female reproduction [61].

This study identified the functional relationships between TTLL polyglutamylases and Nna1 in the nervous and male reproductive systems and revealed that only TTLL1 and TTLL4 generate the Nna1 substrates essential for neuronal degeneration. Further characterization of the substrates may provide insight into whether disrupted polyglutamylation plays a broader role in neurodegeneration and male infertility and present therapeutic opportunities.

## Materials and methods

### Ethics statement

All studies were approved by the St. Jude Children's Research Hospital (SJCRH) Animal Care and Use Committee (ACUC) and complied with the standards set forth in National Institute of Health Guide for the Care and Use of Laboratory Animals (NIH Publication No. 80–23, revised 1996).

### Animals

The *pcd* (Purkinje cell degeneration 3 Jackson:BALB/cByJ-*Agtpbp1*$^{pcd-3J}$/J) mice were purchased from the Jackson Laboratory (Bar Harbor, ME, USA). The *Ttll1*KO (B6.129-Ttll1$^{<tm1Seto>}$/SetoRbrc), *Ttll5*KO (B6.129-Ttll5$^{<tm1Seto>}$/SetoRbrc), *Ttll7*KO (B6;129-Ttll7$^{<tm1Seto>}$/SetoRbrc), and *Ttll11*KO (C57BL/6N-Ttll11$^{<tm1Seto>}$/SetoRbrc) alleles were obtained from Riken BioResource Center (Catalog numbers are RBRC03327, RBRC03328, RBRC03329, and RBRC03337, respectively). A conditional knock-out allele of *Ttll4* (B6Dnk; B6N-Ttll4$^{tm1a(EUCOMM)Wtsi}$/H) was obtained from EMMA (ID: EM:04238), which was crossed with Sox2-Cre (B6.Cg-Edil3$^{Tg(Sox2-cre)1Amc}$/J) transgenic mice (Jackson Laboratory, Stock Number: 008454) to generate a constitutive *Ttll4*KO allele. The strategies to generate KO alleles of all *Ttll* used in the study were summarized in S1 Fig. The *Ttll* and *pcd* double knock-out mice are referred to as *Ttll*KO,*pcd*. Animals were maintained on a 12-h light: 12-h dark cycle with free access to food and water.

### Generation and genotyping of *Ttll*KO and *Ttll*KO,*pcd* double mutants

The heterozygotes of each *Ttll* were bred with *pcd* heterozygotes to generate *Ttll*,*pcd* double heterozygotes, which were further inbred to obtain individual *Ttll*KO,*pcd* double knock-out mice, wild-type littermates, and other intermediate genotypes. Genotyping of *pcd* was done as described previously [16,29]. Genotyping the wild-type and *Ttll*KO alleles was performed by PCR using the primers listed in S1 Table. Genotyping was also confirmed at the RNA level using RT-PCR on total RNA extracted from cerebellum, cerebral cortex, testis, or eye. First strand cDNA was generated using Superscript III kit and amplified with primers that target the deleted region of the respective KO alleles (primers listed in S2 Table). *β-actin* was amplified as a loading control using primers described previously [11].

### qRT-PCR analysis of *Ttll* expression

The levels of *Ttll1-13* and *Agtpbp1* mRNAs from cerebellum, cerebral cortex, olfactory bulb, testis, kidney and liver were evaluated by qRT-PCR. Total RNA was isolated using Trizol (ThermoFisher) and cDNA synthesized using random hexamers and the High-Capacity cDNA Reverse Transcription Kit (Applied Biosystems) according to the manufacturer's instructions. *Ttll1-13* and *Agtpbp1* mouse-specific TaqMan Gene Expression Assays were purchased from ThermoFisher (*Ttll1*, Mm00626796_m1; *Ttll2*, Mm03057809_g1; *Ttll3*, Mm00505799_m1; *Ttll4*, Mm01259116_m1; *Ttll5*, Mm00662072_m1; *Ttll6*, Mm00555632_m1; *Ttll7*, Mm01311964_m1; *Ttll8*, Mm00555808_m1; *Ttll9*,

Mm00512463_m1; *Ttll10*, Mm01263887_m1; *Ttll11*, Mm01336822_m1; *Ttll2*, Mm00812915-m1; *Ttl13*, Mm00625131-m1; and *Agtpbp1*, Mm00508131_m1). qRT-PCR was performed using 2 X TaqMan Fast Reagents Starter Kit (Applied Biosystems) according to the manufacturer's instructions, using the ABI 7900 Fast Real-Time PCR system (Applied Biosystems). PCR conditions were as follows: uracil-N-glycosylase incubation, 50˚C for 2 min, AmpliTaq Gold activation, 95˚C for 10 min, denaturation step 95˚C for 15 s, annealing step 60˚C for 1 min; 40 cycles were performed. Standards for absolute quantification were obtained by cloning *Ttll1-13* and *Agtpbp1* into pcDNA3.1 vector. PCR results were normalized to beta-actin (Actb) [62].

## Histology and immunohistochemistry

The procedures for histological analyses of cerebella, retina, olfactory bulbs, and testes were conducted as described [44,63,64]. A rabbit anti-calbindin D-28K antibody (Chemicon, Temecula, CA, USA) and a rabbit anti-Tbr2 antibody (Abcam, Cambridge, MA, USA) were used at the dilution of 1:500 to visualize Purkinje cells and olfactory bulb mitral cells, respectively and immune complexes were revealed using a peroxidase-conjugated anti-rabbit kit and diaminobenzidine tetrahydrochloride (DAB) substrate (Vector Labs, Burlingame, CA, USA). After immunostaining, sections were counterstained with hematoxylin (Sigma-Aldrich, St Louis, MO, USA).

## Immunofluorescence

**Brain.** Sagittal sections from paraffin embedded mouse brain were cut at a thickness of 5 μm. After deparaffinization, matched sections were heat retrieved in 0.01M sodium citrate buffer (pH 6.0) containing 0.05% Tween-20 (Sigma, St. Louis, MO) for 20 min at 98˚C. After blocking with 10% normal horse serum (Vector Labs, Burlingame, CA) in PBS, sections were incubated overnight at 4˚C with antibodies to Calbindin D-28-K (1:1,000, EMD Millipore, USA), Tbr2 (1:400, Abcam), or GT335 monoclonal antibody (1:1,000, Adipogen, San Diego, CA, USA) or GT335 antibody pre-absorbed with porcine brain tubulin (Cytoskeleton Inc, Denver, CO), then incubated for 1 h at room temperature with Alexa 488-labeled donkey anti-mouse or anti-rabbit antibody (1:200, Invitrogen, San Diego, CA). Sections were counterstained with DAPI (Invitrogen, San Diego, CA).

**Retina.** For 5-month old mice, eyes were dissected, fixed in 4% paraformaldehyde, and embedded in paraffin after sequential dehydration. Immunofluorescence was performed on 5 μm sections. For postnatal day 30 (P30) mice, the retinae were peeled off from eyes and fixed in 4% paraformaldehyde before embedding in OCT embedding medium (Sakura Finetek, Torrance, CA, USA). Immunofluorescence was performed on 12 μm sections. A mouse anti-rhodopsin antibody [41] was used at the dilution of 1:500 to determine the localization of rhodopsin, which is visualized using Alexa Fluor 488 goat anti-mouse IgG (1:500). Images were taken with a Zeiss LSM 710 NLO Confocal Microscope or an Olympus BX60 equipped with DP71 camera. Sections from retina that included the optic nerve were chosen for comparison.

## Protein electrophoresis and immunoblotting

Proteins were separated using a Criterion XT precast gel (4–12% Bis-Tris, (Biorad, Hercules, CA, USA)). After electrophoresis, proteins were transferred onto a nitrocellulose membrane using the Criterion Blotter (Biorad, Hercules, CA, USA). Membranes were incubated with mouse anti-glutamate (GT335, 1:4000, Adipogen), rabbit anti-long-chain polyglutamate (polyE, 1:4,000, Adipogen), or rabbit anti-α-tubulin (EP1332Y, 1:3000, Abcam) antibodies.

Immunoreactivity of proteins was visualized with Supersignal West Pico Chemiluminescence Substrate (Thermo, Rockford, IL, USA) following incubation with HRP-labeled sheep anti-mouse (1:2000, GE Healthcare Sciences, Pittsburgh, USA) or donkey anti-rabbit IgG (1:10,000, GE Healthcare Sciences) antisera.

## Microarray hybridization and analysis

Cerebella of 2-month-old mice were collected and RNA was isolated according to the manufacturers' protocol (TRIzol, ThermoFisher). RNA quality was confirmed by analysis on the Agilent 2100 Bioanalyzer. Total RNA was processed in Hartwell Center for Bioinformatics & Biotechnology at St. Jude Children's Research Hospital according to the Affymetrix GeneChip eukaryote two-cycle target labeling protocol. Biotin-labeled cRNA was added to a hybridization cocktail and then processed automatically on Mouse Genome 2.0_2 or Mouse Clariom S array. Normalized transcript measures were generated from scan intensity files using the RMA algorithm. The array data was quantitated for three Purkinje cell enriched transcripts (Pcp2, Ppp1r17 and Car8) that were identified previously [29] in wild-type; *pcd*; *Ttll1*KO; *Ttll4*KO; *Ttll1*KO,*pcd*, and *Ttll4*KO,*pcd* cerebella.

## Sperm count

Sperm from epididymis and vas deferens were counted using the protocol described previously [65] with minor modifications by including sperm from vas deferens in counting.

## Rota-rod test

To assess motor coordination, balance, and motor learning, gender- and age-matched littermate mice were tested on a rota-rod (San Diego Instruments, San Diego, CA) with an accelerating speed (0 to 40 rpm in 4 min and then hold constant speed for an additional min) as described previously [11,63], and the latency of the mice falling from the rod was scored as an index of their motor coordination. When both locomotor coordination and motor learning were assessed, the test was conducted for 5 consecutive days.

## Statistical methods

The latencies to fall in the rota-rod test were expressed as mean ± SEM (in seconds) and were analyzed for statistical significance using One-way analysis of variance (ANOVA) with repeated measures followed by Bonferroni's multiple comparison test or Student's *t*-test for comparison between samples at the same time point. The level of significance was set at $p < 0.05$. In all other experiments, Student's *t* test was used to compare independent samples for statistical significance. Significance was set at $p$ of $< 0.05$. Student's *t* test was performed using Microsoft Excel software.

## Supporting information

**S1 Table. Primers used for genotyping the knock-out alleles of TTLL.** (DOCX)

**S2 Table. Primers used in RT-PCR to determine the transcripts of TTLL.** (DOCX)

**S1 Data. Numerical data for all figures.** (XLSX)

**S1 Fig. Schematic representation of knock-out strategy of *Ttll* members.** The regions encoding the TTL domain in *Ttll1* (A), *Ttll5* (C), *Ttll7* (D), and *Ttll11* (E) were replaced with a neomycin (neo) selection cassette. (B) A conditional knock-out allele of *Ttll4* where exons 6–8 were flanked with loxp sites was crossed with the maternally expressing *Sox2-cre* transgenic mice to create *Ttll4*KO mice with constitutive deletion of exons 6–8.
(TIF)

**S2 Fig. Loss of *Ttll5, 7*, or *11* did not rescue the locomotor deficit in *pcd* mice.** (A) RT-PCR using primers targeting deleted region in *Ttll5, 7* or *11*KO allele confirmed the absence of *Ttll5, 7*, or *11* transcripts in cerebellum, brain, and testis of respective KO mice. (B) Rota-rod test of 2-month-old gender-balanced littermates of wild-type; *pcd; Ttll5*KO,*pcd;, Ttll7*KO,*pcd;,* and *Ttll11*KO,*pcd* (n = 4–17/genotype) revealed that loss of function of these genes did not improve the locomotor deficit in *pcd* mice. (C) Tubulin polyglutamylation levels in cerebellar lysates from wild-type (WT), *pcd; Ttll5*KO; *Ttll5*KO,*pcd; Ttll7*KO; *Ttll7*KO,*pcd; Ttll11*KO *and Ttll11*KO,*pcd* mice. Loss of function of *Ttll5* and *Ttll11* had little effect on polyglutamylation whereas, loss of function of *Ttll7* in *pcd* markedly reduced GT335 signal, although it did not improve locomotor score.
(TIF)

**S3 Fig. Loss of *Ttll5, 7*, or *11* did not rescue Purkinje cell loss in *pcd* mice.** (A-D) Calbindin D-28K immunofluorescence staining of cerebellar sections from 5-month-old wild-type (A and A'), *Ttll5*KO,*pcd* (B and B'), *Ttll7*KO,*pcd* (C and C') and *Ttll11*KO,*pcd* (D and D') mice. Note the cerebellum of all *Ttll*KO,*pcd* mice strains (B-D) is smaller than that of wild-type (A). (A'-D') Higher magnification of boxed areas in A-D, respectively showed that calbindin-positive Purkinje neurons are not restored in *Ttll5*KO,*pcd, Ttll7*KO,*pcd or Ttll11*KO,*pcd* mice. ML: Molecular Layer; PCL: Purkinje Cell Layer; GCL: Granule Cell Layer.
(TIF)

**S4 Fig. GT335 antibody specificity.** Sections of adult cerebellum were immunostained with GT335 antibody (green) without (A) or with (B) porcine tubulin pre-absorption and nuclei visualized with DAPI (blue). Note large reduction in immunoreactive signal with the pre-absorbed antibody.
(TIF)

**S5 Fig. Knocking out *Ttll5, 7*, or *11* did not rescue olfactory bulb mitral cell degeneration in *pcd* mice.** Sections of olfactory bulbs from 5-month-old wild-type (A, A', E), *Ttll5*KO,*pcd* (B, B', and F), *Ttll7*KO,*pcd* (C, C', and G), and *Ttll11*KO,*pcd* (D, D', and H) mice were immunostained for Tbr2, which recognizes mitral cells and tufted cells located in 2 distinct layers. In all genotypes, Tbr2-positive cells are present in the outer layer where tufted cells are located (A'-D'), whereas Tbr2-positive mitral cells are present in wild-type (A'), but almost completely absent in *Ttll5*KO, *pcd* (B'), *Ttll7*KO,*pcd* (C'), and *Ttll*11KO,*pcd* (D') mice. (E-H) Bright-field immunohistochemistry images showed that Tbr2-positive mitral cells are present in wild-type mice (E), but largely absent in *Ttll5*KO,*pcd* (F), *Ttll7*KO,*pcd* (G), or *Ttll11*KO,*pcd* (H) double mutants. Arrows indicate Tbr2-positive cells in the MCL and dotted red lines indicate position of MCL. GL: Glomerular Layer; EPL: External Plexiform Layer; MCL: Mitral Cell Layer; GCL: Granule Cell Layer.
(TIF)

## Acknowledgments

We thank Dr. Michael Dyer from the Department of Developmental Neurobiology at St. Jude Children's Research Hospital (SJCRH) for advice and assistance on experiments with retinae and for providing rhodopsin antibody.

## Author Contributions

**Conceptualization:** Hui-Yuan Wu, James I. Morgan.

**Data curation:** Hui-Yuan Wu, Yongqi Rong, Parmil K. Bansal, Peng Wei, Hong Guo, James I. Morgan.

**Formal analysis:** Hui-Yuan Wu, Yongqi Rong, Parmil K. Bansal, Peng Wei, James I. Morgan.

**Funding acquisition:** James I. Morgan.

**Investigation:** Hui-Yuan Wu, Yongqi Rong, Parmil K. Bansal, Peng Wei, Hong Guo, James I. Morgan.

**Methodology:** Hui-Yuan Wu, Yongqi Rong, Parmil K. Bansal, Peng Wei.

**Project administration:** James I. Morgan.

**Supervision:** James I. Morgan.

**Validation:** Hui-Yuan Wu, Yongqi Rong, Peng Wei, James I. Morgan.

**Visualization:** Hui-Yuan Wu, Yongqi Rong, James I. Morgan.

**Writing – original draft:** Hui-Yuan Wu, Yongqi Rong, Parmil K. Bansal, Peng Wei, James I. Morgan.

**Writing – review & editing:** Hui-Yuan Wu, Yongqi Rong, Parmil K. Bansal, Peng Wei, James I. Morgan.

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
