## [Decision Letter · Decision Letter 0]

28 Apr 2021

Dear Dr Rong,

Thank you very much for submitting your Research Article entitled 'TTLL1 and TTLL4 polyglutamylases are required for the neurodegenerative phenotypes in pcd mice' to PLOS Genetics.

The manuscript was fully evaluated at the editorial level and by independent peer reviewers. The reviewers appreciated the attention to an important problem, but raised some substantial concerns about the current manuscript. Based on the reviews, we will not be able to accept this version of the manuscript, but we would be willing to review a much-revised version. We cannot, of course, promise publication at that time.

Each of these expert reviewers thought your manuscript holds broad interest and potentially significant impact, but as per several concerns it seems that a key take-home message - that TTLL4 but not the other paralogues rescues the CCP1 ko phenotype - was significantly missed or lost, due to a number of factors including data presentation and analysis that falls short, and unclear writing/organization. In addition, as Reviewer 1 noted, the mere fact that TTLL6 and 11 are not expressed highly seems to undermine the observation of non-rescue by these genes. As a result, you will need to make significant revisions - satisfying the reviewers with a combination of revised writing/emphasis, new data or new data analysis - before the manuscript can be considered further. While some of the concerns are more important than others, we will require a detailed list of your responses to the review comments and a description of the changes you have made in the manuscript as we will put any revised manuscript back in their hands.

If you decide to revise the manuscript for further consideration at PLOS Genetics, please aim to resubmit within the next 60 days, unless it will take extra time to address the concerns of the reviewers, in which case we would appreciate an expected resubmission date by email to plosgenetics@plos.org.

[LINK]

We are sorry that we cannot be more positive about your manuscript at this stage. Please do not hesitate to contact us if you have any concerns or questions.

Yours sincerely,

Wayne N. Frankel

Associate Editor

PLOS Genetics

Gregory Barsh

Editor-in-Chief

PLOS Genetics

Reviewer's Responses to Questions

**Comments to the Authors:**

Reviewer #1: The levels of polyglutamylation have been shown to be critical for neuronal survival and function. Loss of CCP1 (as in pcd mice), an enzyme that removes this modification, results in the degeneration of select neuronal populations in mice, indicating that hyperglutamylation is particularly problematic. Previous studies have shown that loss of TTLL1, a tubulin tyrosine ligase-like enzyme, which adds glutamate residues to substrate proteins can protect Purkinje cells in pcd mice. Here, Wu et al. expand on these findings, investigating the contribution of TTLL1 to degeneration of other neuronal populations in pcd mice. They also examine the contribution of 5 other members of the tubulin tyrosine family to phenotypes in pcd mice. This systematic genetic analysis can shed light on the contribution of these different enzymes to cellular homeostasis and survival and is thus of interest to the readers of Plos Genetics. However, there are a number of concerns that must be addressed before the paper is suitable for publication.

Major concerns:

1. The authors show that loss of TTLL6 and 11 does not impact Purkinje cell degeneration in pcd mice. However, these genes seem to be barely expressed in Purkinje cells (based on the Allen brain atlas and for TTLL6, the authors’ qPCR data) and thus the inability to rescue this pcd phenotype is likely reflective simply of the spatial or cell type differences in expression rather than functional differences, such as differences in substrate specificity. It is possible that these genes may be able to functionally compensate/act like TTLL1 if they were actually expressed in these cells. Similarly, the differential modulation of the olfactory bulb and retinal phenotype may also simply be reflective of the expression level in these cell types rather than differential substrate specificity. The authors should modify their discussion and results section to clarify this, as it may mislead readers.

An interesting experiment, though probably beyond the scope of this study, is to investigate whether transgenic overexpression of TTLL6/11 in Purkinje cells can modulate the phenotype of the pcd/TTLL1 double mutants.

2. As Purkinje cells are a small fraction of the cells in the cerebellum (about 0.3%), qPCR or western blots on whole cerebellar lysate (consisting mostly of granule cells) do not accurately represent molecular changes in these cells. This makes it difficult to correlate molecular changes to phenotypes, for instance the western blots of glutamylation levels in pcd/ttll4 double mutants to the rescue of Purkinje cell loss. The inclusion of immunohistochemistry or in situ hybridization (e.g. RNAScope) for the different members of this family would be helpful for the interpretation of such results. In addition, (semi)-quantitative immunohistochemistry for tubulin glutamylation in the cerebellum would be helpful to determine the role of tubulin polyglutamylation in the neurodegenerative phenotype.

3. The inclusion of microarray data from a timepoint at which most Purkinje cells have already been lost in the pcd mouse adds little meaningful data beyond that shown by the histology as gene expression changes are reflective of the absence of these cells.

Minor suggestions:

1. It is unclear which lobule is shown in the cerebellar histology, or even if the same lobule is shown for the different genotypes (Fig. 2C, 3D etc). This should be clarified, and if possible, a low magnification image of the entire cerebellum should be included.

2. Quantification/statistical analysis should be included for the western blots. In addition, some of the western blots including Fig. 3B and 4B have saturated signal/shadow bands.

3. Do TTLL6/11 affect testes function/male infertility?

4. The authors mention that loss of function mutations in TTLL5 have been shown to cause photoreceptor loss in mice (line 485). Has photoreceptor loss indeed been observed in mouse models? If so, can the authors discuss the discrepancy with their data?

Reviewer #2: In their study, Wu et al. investigate the recently discovered mechanism of hyperglutamylation-induced neurodegeneration. Polyglutamylation is a posttranslational modification (PTM) of tubulin that is highly enriched on microtubules in neurons. Recent work had shown that when this PTM is abnormally accumulated due to the absence of the deglutamylating enzyme CCP1, neurons degenerate. The most emblematic neuron type to degenerate are the cerebellar Purkinje cells.

It had also been shown by several teams that removing one of the key glutamylase enzymes, TTLL1, in the CCP1-KO background prevented the degeneration of the Purkinje cells, demonstrating the causative link between accumulation of polyglutamylation and degeneration.

The question that had so far not been answered is whether other polyglutamylating enzymes are also involved in the degeneration observed in CCP1-KO mice. The current manuscript addresses this question in a systematic manner.

The authors generated knockout mouse models for 6 additional TTLL glutamylase enzymes, and systemically crossed them with the CCP1-KO (pcd) mouse to test their impact on Purkinje-cell degeneration. They find that, as previously shown, TTLL1-KO rescues this phenotype, but rather unexpectedly also TTLL4-KO did so. By contrast, none of the other TTLL enzymes could rescue Purkinje cell degeneration, which was surprising given that some of them are highly expressed in the brain.

Following these exciting results, the authors further show that two other degenerative phenotypes of pcd mice, the degeneration of Mitral cells in the olfactory bulb and the degeneration of the retina are also prevented by knockout of TTLL1 or TTLL4, however, the retina degeneration is only partially protected by TTLL4-KO. By contrast, the male sterility of CCP1-KO mice is not reverted when combined with TTLL1- nor TTLL4-KO.

The crux of these findings is the question why TTLL4-KO protects similar to TTLL1-KO, while other TTLLs have no effect. Most importantly, polyglutamylation, in CCP1-TTLL4-dKO mice is not reduced, while this is the case in CCP1-TTLL1-dKO mice.

The authors have generated an impressive number of combinatorial mouse models to demonstrate the role of each individual TTLL enzyme, which is a remarkable tour de force. Moreover, their finding that only one other glutamylating enzyme, TTLL4 protects neurons from degeneration similar to what had previously shown for TTLL1, while all other enzymes of this family, even those that are highly expressed in the brain, had no effect, is novel, important and exciting. It is therefore unfortunate the in the current form, the manuscript does not value this finding sufficiently. The key problem of this manuscript is its blunt enumeration of experiments the authors have performed, the lack of effort to make figures easy to understand, to guide the reader through their work. Moreover, no attempts to provide quantitative measures for phenotypes have been made, which would be particularly important for the TTLL4/CCP1-dKO model (see below). In its current form, the paper is difficult to read and there are a number of open questions that must be addressed before considering it for publication.

Major points:

1) Fig. 1A: First of all, why did the authors omit TTLL2, TTLL9 and TTLL13 which have been predicted / demonstrated to be glutamylases?

However, the main problem with this figure and the conclusions drawn from it is the fact that it is very difficult to conclude the roles of these enzymes in the tested tissues from these analyses. How can Q-PCR data from different genes be compared? Most likely not at all. What can, however, be compared is the relative expression of the same gene in different organs or tissues. For TTLL6, for instance, it is obvious that it might not play a role in the nervous system, as its expression levels are very low compared to testes. All other enzymes show similar expression levels in brain and testes. To make a convincing point about the specific expression of the enzymes in the nervous system, the authors must compare their expression levels in the brain to the expression levels in other organs, and preferably organs where polyglutamylation plays a less important role (testes are surely not the best control here). Without this comparison, it is hard to tell if the numbers the authors obtained stand for a strong or weak expression.

Finally, it is not clear how many times these analyses were repeated (no statistical analysis of the data). Also, representing the Q-PCR data in the form of diagrams would improve the readability of the figure.

2) Quality of data representation: The paper has some major problems with the quality and the readability of the figures. This is not only a problem for the readability of the manuscript, it might also affect the interpretation of the results. The two main concerns are:

2a) Immunoblots lack molecular weight markers, and are very closely cropped. This is not good practise.

2b) The authors never comment whether they can distinguish alpha- and beta-tubulin in their blots, which would be helpful as they interpret a lot of their results in the light of the enzymatic specificities of the enzymes.

2c) All polyE blots in this manuscript appear to be overloaded, and can therefore not reveal subtle differences in polyglutamylation levels that might occur in some cases. While the previously reported very strong difference between wild type and CCP1-KO (pcd) is visible under these experimental conditions, the authors might have missed a more subtle decrease in polyE signal, for instance in CCP1/TTLL4 dKO mice that show partial Purkinje-cell survival. Being able to see such subtle differences would completely change the interpretation of the paper.

2d) Histology images are not homogenously represented. First, they are so different in terms of colour, and region selected it is not obvious for a reader that the brain regions depicted are comparable. This concerns Fig. 2C, 3D, 4C and 6 (for instance, compare panels in Fig. 2C – they all look completely different). Images need to be better processed, comparable regions should be shown. Brain layers should be labelled. Moreover, given that brain histology is basically the key readout of this work, entire cerebella should be shown additionally to the zoomed-in region.

3) It is not clear why the authors first analyse the TTLL6-KO mice, having shown in Fig. 1A that this enzyme is not expressed in the nervous system? This is counter-intuitive, and becomes particularly perturbing at line 202, where the authors simply conclude that TTLL1, TTLL6 and TTLL11 elongate polyglutamylation on alpha-tubulin, but only TTLL1 appears to contribute to the degeneration of Purkinje cells – citing Fig. 1A. Going to Fig. 1A now, the reader discovers that TTLL6 is not expressed in cerebellum (making the lack of rescue obvious), but TTLL11 is Q-PCR-wise more than 2x more expressed than TTLL1. So how can this be explained? This most likely needs to be put in the context of what Q-PCR data really tell us (see point 1) above).

4) The most striking discovery of this paper is the (partial) rescue the CCP1-related phenotypes by knockout of TTLL4. However, it raises the important question of the underlying mechanism given that in the western blots it seems that TTLL4-KO does not reduce hyperglutamylation, while TTLL1-KO does. As mentioned in point 2c), there might be a slight reduction of polyglutamylation the authors have not detected because of the way they perform their blots, but the question still remains of how TTLL4-KO can rescue given that is does not lower polyglutamylation as does TTLL1-KO. One possible answer could be that in fact all of the observed rescue phenotypes are partial.

Indeed, the authors show that TTLL4-KO can only partially rescue the CCP1-induced degeneration of the retina, but they claim at the same time that it can fully rescue mitral and Purkinje cells based on snapshots of olfactory bulb and cerebellum without any quantification or multiple experiments. Given that in mouse models in which Purkinje cells degenerate only partially, this partial degermation is not homogenous throughout the cerebellum (e.g. Fig. 1G in Liu Y, Lee JW, Ackerman SL (2015) Mutations in the Microtubule-Associated Protein 1A (Map1a) Gene Cause Purkinje Cell Degeneration. J Neurosci 35: 4587-4598), it would be first of all essential to show images of the entire cerebellum (and similarly, the entire olfactory bulb) in order to show that the region the authors have presented currently is representative. Moreover, the authors might attempt to quantify the number of Purkinje cells (and mitral cells) in the different mouse models to determine whether there is a partial degeneration similar to what they see in the retina. Finally, it might be worth checking whether the protective effect of TTLL4 is as efficient as the one by TTLL1 by investigating older mice.

5) The gene expression study in Fig. 5 is awkward: all the genes that the authors find to be downregulated are Purkinje-cell specific genes. So obviously, in all mouse models where the Purkinje cells are lost, the genes specifically expressed in these cells cannot be detected anymore. But this does by no means prove their downregulation, but simply mirrors the loss of Purkinje cells.

Minor points:

1) In the abstract, the authors write “Genetic disruption of polyglutamylation causes neurodegenerative phenotype”. This is misleading, as it could be interpreted as loss of glutamylation (“disruption”). However, what really happens is that accumulating polyglutamylation leads to degeneration. This should be re-formulated.

2) The authors omit the notion that while all glutamylases have a previously demonstrated preference for initiation or elongation, as well as for alpha- or beta-tubulin, these are preferences and not exclusive activities. In other words, one cannot exclude that TTLL1 initiates the glutamate chains it later elongates. They should consider the potential promiscuity of the enzymes when discussing their findings.

3) Fig. 1B: it is important to introduce the reader to polyglutamylation and the involved enzymes. However, this could be done a bit more intuitive than in the current figure that oversimplifies a lot. Why not introducing the notion of alpha- and beta-tubulin preferences into this schematic, and remove it from Fig. 1A, where data are shown?

4) line 182: the authors must explain the connection between PGs1 and TTLL1 mutant mice.

5) in Fig. 7, the zoom does not really reveal what the authors want to show. A more explicit representation should be sought.

6) in Fig. 8, single-channel images as in Fig. 7 would be useful to clearly see the phenotype the authors want to point out. Fig. 7 and 8 could be combined.

7) In the discussion, the authors propose that their observation of the importance of TTLL4 for Purkinje cells might be explained by both, alpha- and beta-tubulin glutamylation participating in this phenotype. However, they did not show that TTLL4 is contributing to beta-tubulin glutamylation in neurons or brain tissue, and they also do not discuss why in this case TTLL7, a known beta-tubulin glutamylase with a very strong expression in brain, and important functions in neurons (Ikegami et al., 2006) does not rescue CCP1-mediated degeneration.

**Have all data underlying the figures and results presented in the manuscript been provided?**

Reviewer #1: **No: **No GEO accession number for microarray dataset provided

Reviewer #2: Yes

PLOS authors have the option to publish the peer review history of their article (what does this mean?). If published, this will include your full peer review and any attached files.

Reviewer #1: No

Reviewer #2: No

---

## [Decision Letter · Decision Letter 1]

17 Feb 2022

Dear Dr Rong,

Thank you very much for submitting your Research Article entitled 'TTLL1 and TTLL4 polyglutamylases are required for the neurodegenerative phenotypes in pcd mice' to PLOS Genetics.

The manuscript was reviewed again by the original reviewers.  Reviewer #1 was satisfied with the revision, as was, generally, Reviewer #2 but this reviewer identified some aspects of the presentation that need improvement before we can proceed further, including noting some errors as well as suggestions for significantly clarifying the presentation and writing.  We would ask that you look through these suggestions very carefully and ideally make most or all of the changes as we agree it would improve the presentation and thus the impact.

[LINK]

Yours sincerely,

Wayne N. Frankel

Associate Editor

PLOS Genetics

Gregory Barsh

Editor-in-Chief

PLOS Genetics

Reviewer's Responses to Questions

**Comments to the Authors:**

Reviewer #1: The authors have addressed my concerns, and the manuscript is much improved. I now support its publication. I would suggest that the authors label the relevant genotypes on the panels of their immunofluorescence figures (Fig3,4,6, 7-9) to make things easier for readers, and avoid labels such as panel Aa''.

Reviewer #2: The authors present a thoroughly revised manuscripts in which they carefully addressed all referees’ comments. As a result, the revised manuscript has hugely improved. Most importantly, the authors have performed a series of new experiments to prove the reproducibility of their observations, which was one of the key concerns in the first round of review. They have further re-focused the story onto the key findings that TTLL1 and TTLL4, but none of the other tested TTLL enzymes play a role in neurodegeneration induced by accumulation of polyglutamylation. This improves the logical flow of the new manuscript, while all important information is still available in the supplementary material.

Given the solidness of this work, it will be a strong contribution to the further mechanistic understanding of the newly emerging role of the tubulin modification polyglutamylation in neurodegeneration. In particular, the observation of the authors that the enzyme TTLL4 contributes to this novel form of degeneration despite the fact that TTLL4 is not the most-strongly expressed TTLL in the brain is intriguing and will help to further elucidate the potentially complex mechanisms leading to the loss of neurons in the future.

Overall, the revised manuscript shows no major weaknesses, experimental data are solid, and results are meticulously discussed. A couple of minor concerns should be addressed before considering the paper for publication:

1) The authors should carefully re-read the manuscript. It appears that there are parts that have not been fully revised, as for instance some figure numbers are wrong: in the discussion they cite Fig 9 for testes, whereas this is Fig 10 in the revised manuscript.

2) In the introduction (first sentence) the authors write that polyglutamylation is involved in neurodegeneration. This is a bold statement, and could mislead the reader into thinking of common neurodegenerative disorders such as Alzheimer’s disease. Specifying that polyglutamylation is involved in a rare condition of childhood degeneration would be more appropriate.

3) In Fig 4A, panel a, the axon tracts stained with Calbindin are less prominent in wildtype as compared to the mutant in panel b. This is most likely related to the choice of the brain section, but could give the misleading impression that wildtype Purkinje cells have a smaller axon tract as compared to the mutant. The authors might consider replacing the image.

4) What makes the manuscript really difficult to read are the figures. There are a number of style issues with the figures, which the authors might want to address:

• Fig 1B: a table packed with so many numbers is far from being easy to read. Why don’t the authors choose to represent their data as graphs?

• Fig 3, 4A, 6, 7, 8, 9 – without a labelling of the panels (genotypes, antibodies used, etc.) these figures are very hard to decipher (one goes forth and back between figure and legend many times before being sure to correctly understand the data). Given that the authors have shown willingness to put such labels in Fig 10, they might want to consider doing so consistently in all figures.

• In most figures showing immunohistochemistry, the authors wrote text labels over the figure panels. Most of these labels are very small and hard to read, and sometimes it is even impossible to read them because of the image below. Bold letters, and perhaps some background shading, could help here. Scale bars are also very thin lines, thus hard to spot. Given that different magnifications are shown side-by-side, it would not heart to directly label the scale bars in the panels.

• Fig 3, 4A and 6 are monochrome images: why do the authors show them in green? They would be much more contrasted in black and white.

• To make Fig 7 and 8 more readable for the non-specialist, a better labelling of the layers, addition of zoom images to show details, plus a schematic representation of the olfactory bulb could be very helpful.

• Fig 2C,D – the authors plot 4 different measurements using very hard-to-distinguish symbols. They might consider the use of color to make these graphs easier to read.

• Fig 4B, 5C,D: the text in these graphs is much too small to be readable.

• Some of the panels showing immunohistochemistry images are very small, in particular Fig 3 and 8. Given the wealth of important information on those images, they could be much larger than for instance an immunoblot with only 4 protein bands in Fig 5.

**Have all data underlying the figures and results presented in the manuscript been provided?**

Reviewer #1: Yes

Reviewer #2: **No: **If I am not mistaken, there are no numerical data to the graphs shown in the manuscript, please verify.

PLOS authors have the option to publish the peer review history of their article (what does this mean?). If published, this will include your full peer review and any attached files.

Reviewer #1: No

Reviewer #2: No

---

## [Editor Report · Decision Letter 2]

24 Feb 2022

Dear Dr Rong,

Thank you very much for submitting your Research Article entitled 'TTLL1 and TTLL4 polyglutamylases are required for the neurodegenerative phenotypes in pcd mice' to PLOS Genetics.

We reviewed your revision and we agree that most of the changes you made are responsive and acceptable.  But in the interest of readability, as the reviewer had been concerned, there are still a few minor changes that we would ask that you make before the manuscript is formally accepted.  Once you address these we will be poised to make the editorial decision very quickly.

First, in principle it is okay if you leave Figure 1b in table form, but it is still an eyeful to look at. Since the gene expression values are whole integers with no decimals, the second decimal place in the SEM is unnecessary and clutters up the table – please use only one decimal place for the SEM values.  Also, please put a space between the expression value and the ± symbol.

Your including genotypes in the figures as the reviewer asked is helpful, but it needs to be clearer still.  It is acceptable to use shorthand gene symbols in the genotypes rather than full formal symbols in the figures, as long as the shorthand is defined somewhere clearly in the text (e.g. Methods).  Thus, pcd should be defined as Agtpbp1 (italicized) pcd (superscript, non italicized, no spaces), and then if you mean homozygous spell that out too (e.g. Agtpbp1 pcd/Agtpbp1 pcd or heterozygous (Agtpbp1 pcd/Agtpbp1 +).  Similar for Ttl1 KO (there is probably an official gene symbol for this KO allele – you can check with JAX if one has been registered; if not, you can use the minus sign to indicate ko, i.e. -).  Also, for pcd specifically in some figures you use pcd and others pcd^3J; please be consistent if you only used one of these.  Last, the double mutant shorthanding is confusing because you are using a slash which is typically reserved for use in genotype copy number; instead of the slash, to indicate double mutants you should use a comma.  That is, if you are going to shorthand Ttl1 mice as Ttl1KO and Agtpbp1 as pcd, and specify in the Methods that these are homozygous, then you could use in the shorthand for the figure: Ttl1KO, pcd).  If there are mixed copy numbers, then that could be further specified, e.g. Ttl1KO/+, pcd/pcd.

Again, once these issues are addressed we will finalize a decision rapidly.

[LINK]

Yours sincerely,

Wayne N. Frankel

Associate Editor

PLOS Genetics

Gregory Barsh

Editor-in-Chief

PLOS Genetics

---

## [Editor Report · Decision Letter 3]

14 Mar 2022

Dear Dr Rong,

We are pleased to inform you that your manuscript entitled "TTLL1 and TTLL4 polyglutamylases are required for the neurodegenerative phenotypes in pcd mice" has been editorially accepted for publication in PLOS Genetics. Congratulations!

Yours sincerely,

Wayne N. Frankel

Associate Editor

PLOS Genetics

Gregory Barsh

Editor-in-Chief

PLOS Genetics

Comments from the reviewers (if applicable):

**Data Deposition**

http://datadryad.org/submit?journalID=pgenetics&manu=PGENETICS-D-21-00374R3

**Press Queries**

---

## [Editor Report · Acceptance letter]

8 Apr 2022

PGENETICS-D-21-00374R3 

TTLL1 and TTLL4 polyglutamylases are required for the neurodegenerative phenotypes in pcd mice 

Dear Dr Morgan, 

We are pleased to inform you that your manuscript entitled "TTLL1 and TTLL4 polyglutamylases are required for the neurodegenerative phenotypes in pcd mice" has been formally accepted for publication in PLOS Genetics! Your manuscript is now with our production department and you will be notified of the publication date in due course.

With kind regards,

Agnes Pap

PLOS Genetics

On behalf of:
